



# Anticipating CRISTAL: An exploration of multi-frequency satellite altimeter snow depth estimates over Arctic sea ice, 2018-2023

Jack C. Landy[1], Claude de Rijke-Thomas[2], Carmen Nab[3,4], Isobel Lawrence[5], Isolde A. Glissenaar[2], Robbie D.C. Mallett[1], Renée M. Fredensborg Hansen[6,7,8], Alek Petty[9], Michel Tsamados[3], Amy R. Macfarlane[1,10], and Anne Braakmann-Folgmann[1]

[1]UiT The Arctic University of Norway, Department of Physics and Technology, PO Box 6050 Langnes, 9037 Tromsø, Norway
[2]University of Bristol, School of Geographical Sciences, Bristol BS8 1SS, UK
[3]Centre for Polar Observation and Modelling, Department of Earth Sciences, University College London, UK
[4]Ocean Forecasting Research & Development, Met Office, UK
[5]ESA ESRIN, Frascati, Italy
[6]DTU Space, National Space Institute, Technical University of Denmark (DTU), Department of Geodesy and Earth Observation, Kgs. Lyngby, Denmark
[7]Norwegian University of Science and Technology (NTNU), Department of Civil and Environmental Engineering, Trondheim, Norway
[8]Department of Arctic Geophysics, The University Centre in Svalbard (UNIS), Longyearbyen, Norway
[9]Earth System Science Interdisciplinary Center, University of Maryland, College Park, MD, 20740
[10]Northumbria University, Ellison Pl, Newcastle upon Tyne NE1 8ST, UK

**Correspondence:** Jack Landy (jack.c.landy@uit.no)

**Abstract.**

The EU and ESA plan to launch a dual-frequency Ku- and Ka-band polar-orbiting synthetic aperture radar (SAR) altimeter, CRISTAL (Copernicus Polar Ice and Snow Topography Altimeter), by 2028 to monitor polar sea ice thickness and its overlying snow depth, among other applications. However, the interactions of Ku- and Ka-band radar waves with snow and sea ice are not fully understood, demanding further research effort before we can take full advantage of the CRISTAL observations. Here, we use three ongoing altimetry missions to mimic the sensing configuration of CRISTAL over Arctic sea ice and investigate the derived snow depth estimates obtained from dual-frequency altimetry. We apply a physical model for the backscattered radar altimeter echo over sea ice to CryoSat-2's Ku-band altimeter in SAR mode and to the SARAL mission's AltiKa Ka-band altimeter in low-resolution mode (LRM), then compare to reference laser altimetry observations from ICESat-2. ICESat-2 snow freeboards (snow + sea ice) are representative of the air-snow interface, whereas the radar freeboards of AltiKa are expected to represent a height at or close to the air-snow interface, and CryoSat-2 radar freeboards a height at or close to the snow-ice interface. The freeboards from AltiKa and ICESat-2 show similar patterns and distributions; however, the AltiKa freeboards do not thicken at the same rate over winter, implying that Ka-band height estimates can be biased low by 10 cm relative to the snow surface due to uncertain penetration over first-year ice in spring. Previously-observed mismatches between radar freeboards and independent airborne reference data have been frequently attributed to radar penetration biases, but can be significantly reduced by accounting for surface topography when retracking the radar waveforms. Waveform simulations of CRISTAL in its expected sea ice mode reveal that the heights of the detected snow and ice interfaces are more sensitive



to multi-scale surface roughness than snow properties. For late-winter conditions, the simulations suggest that the CRISTAL Ku-band radar retrievals will track a median elevation 3% above the snow-ice interface, because the radar return is dominated
by surface scattering from the snow-ice interface which has a consistently smoother footprint-scale slope distribution than the air-snow interface. Significantly more backscatter is simulated to return from the air-snow interface and snow volume at Ka-band, with the radar retrievals tracking a median elevation 10% below the air-snow interface. These model results generally agree with the derived satellite radar freeboards, which are consistently thicker for AltiKa than CryoSat-2, across all measured snow and sea ice conditions.



## 1 Introduction

The Arctic marine system is one of the fastest changing environments on Earth. Since the 1980s the area of Arctic sea ice at the end of the summer melting season has approximately halved (Stroeve and Notz, 2018) while around three-quarters of the ice volume has disappeared (Kwok, 2018). The latest synthesis of climate model projections in the IPCC's Sixth Assessment Report suggests at least one practically ice-free summer in the Arctic is likely before 2050, regardless of the $CO_2$ emission

scenario (Fox-Kemper et al., 2021). However, the spread in the projected timing of a regularly ice-free Arctic spans more than 30 years (Notz and Community, 2020). Much of this uncertainty comes from the structure of the sea ice component of the climate model (Bonan et al., 2021), and the Coupled Model Intercomparison Project (CMIP) Phase 6 ensemble do not reproduce the observed patterns of pan-Arctic sea ice thickness (SIT) accurately (Watts et al., 2021). According to Massonnet et al. (2018), the "current main obstacle to reducing uncertainties in projected sea-ice volume or area trends is not the complexity

of the models used, but rather the lack of long-term and reliable estimates of sea-ice volume that can be used to constrain their projections." Upgrading the observational SIT record will benefit climate model projections, improve the initialization of seasonal sea ice forecasts (e.g., Bushuk et al., 2017), and provide enhanced understanding of the Arctic's fast-changing sea ice mass and energy budgets.

The EU and ESA plan to launch a new Sentinel Expansion Mission: the Copernicus Polar Ice and Snow Topography Altimeter (CRISTAL) by 2028 to continue and enhance the record of spaceborne sea ice thickness observations in the Arctic (Kern et al., 2020). The observations from CRISTAL will build on a legacy of pan-Arctic SIT generated from the polar-orbiting ESA CryoSat-2 mission, since 2010 (e.g., Landy et al., 2022), and from historic radar altimeters (ERS-1/-2 and ENVISAT RA-2) with sub-Arctic coverage since 1995 (e.g., Bocquet et al., 2023). Complementary observations of pan-Arctic sea ice thickness

have also been produced from the polar-orbiting NASA ICESat (2003-2009) and ICESat-2 (2018-onwards) spaceborne laser altimetry missions (Kwok and Cunningham, 2008; Petty et al., 2023a). CRISTAL will carry a dual-frequency interferometric Ku- and Ka-band SAR altimeter, with the goal to produce profiles of SIT at ∼250 m intervals along the track of the satellite with conventional delay-Doppler processing and <80 m with Fully-Focused processing (Kern et al., 2020). The dual-frequency sensor is motivated by the assumption that Ku-band pulses penetrate through the snow layer on sea ice (e.g. Beaven et al.,

1995), whereas Ka-band pulses scatter at the upper snow surface or layer (Guerreiro et al., 2016). This allows snow depth to be estimated from the height difference of the backscattered echoes. One of the largest sources of uncertainty in state-of-the-art SIT datasets comes from the snow load, which is conventionally obtained from a climatology (e.g., Warren et al., 1999) or estimated from an external source (such as fused climatology and passive-microwave-derived estimates or reanalysis-based accumulation models) to convert the altimeter's freeboard measurement to an estimate of SIT (Tilling et al., 2018; Mallett et al.,

2021; Glissenaar et al., 2021). If the snow depth can be accurately measured by CRISTAL, concurrently with the measured sea ice freeboard, then uncertainty in the derived SIT may start approaching the 0.15 m @ 25-km length scale targeted by the mission (Kern et al., 2020, (compared to the typical >0.5 m uncertainty on CryoSat-2 SIT estimates)).



The basic assumptions of radar backscatter that have motivated the configuration of the dual-frequency CRISTAL payload
are significantly more complex in reality. A variety of studies based on theoretical methods (Nandan et al., 2017, 2020; Tonboe
et al., 2021; Meloche et al., 2024), in-situ (or surface-based) radar (Willatt et al., 2009; Stroeve et al., 2020; Nandan et al.,
2023; Willatt et al., 2023), airborne radar (Willatt et al., 2011; King et al., 2018; de Rijke-Thomas et al., 2023), and satellite
radar observations (Ricker et al., 2015; King et al., 2015; Nab et al., 2023), have challenged the assumption that spaceborne
Ku-band radar altimetry (e.g., CryoSat-2) can accurately detect the height of the snow-ice interface over snow-covered sea
ice floes. It has been suggested that CryoSat-2 may be sensitive to a radar scattering distribution with mean height within the
snowpack, 60-90% deep in the snow relative to the air-snow interface (Armitage and Ridout, 2015; Lawrence et al., 2018; Nab
et al., 2023; Landy et al., 2022), implying that CryoSat-2 radar freeboards are consistently biased thick if there are not strong
competing biases. Moreover, studies investigating coincident satellite observations from CryoSat-2 and the CNES/ISRO Ka-
band altimeter mission SARAL AltiKa have challenged the assumption that Ka-band radar freeboards accurately represent the
height of the air-snow interface over sea ice (Armitage and Ridout, 2015; Lawrence et al., 2018). AltiKa may also, therefore,
be sensitive to a radar scattering distribution with a mean height within the snowpack of 0-40% of the snow depth (Guerreiro
et al., 2016; Armitage and Ridout, 2015; Nab et al., 2023).

Radar pulse propagation at Ku- and Ka-band depends on the electromagnetic characteristics of the snow and sea ice cover.
The radar wave can be scattered at the interfaces between air and snow, and snow and ice, depending on the dielectric contrast
between layers and the "radar scale" mm-cm roughness of the interface, i.e. similar to the wavelength: ∼2 cm at Ku-band and
∼8 mm at Ka-band (Kurtz et al., 2014; Kwok, 2014; Landy et al., 2019; de Rijke-Thomas et al., 2023; Meloche et al., 2024).
The wave can also be scattered and absorbed within the snow volume, depending on the snow density, grain size, dielectric
properties (moisture, related to brine content) and structure (wind crusts, ice lenses) (Nandan et al., 2017, 2023; Willatt et al.,
2023). These surface and volume scatterers are then distributed over a range of heights, with respect to the spherical wavefront
of the propagating pulse, by the large-scale (1-1000s m scale) topography of the sea ice cover. A rougher surface topography
produces a broader backscattered radar echo (Kurtz et al., 2014; Landy et al., 2020). This complex set of scattering mechanisms
can produce a mean height of the backscattered radar intensity that does not exactly match the absolute mean height of the
snow surface (at Ka-band) or sea ice surface (at Ku-band), even in the case where the waves negligibly or totally penetrate the
snowpack. Ka-band waves are theoretically 1-2 orders of magnitude more sensitive to snow volume scattering than Ku-band
waves (Mätzler, 1998; Rémy et al., 2015), and AltiKa freeboards are generally thicker than CryoSat-2 freeboards (Armitage and
Ridout, 2015); however, the relative radar penetration depth into snow may not always be 0% at Ka-band and 100% at Ku-band.

It is not only the geophysical properties of the target that affect the returning altimeter waveform. The sensor design and
measurement geometry can affect the relative importance of each scattering mechanism, and different methods for interpret-
ing the radar signal can impact derived geophysical parameters such as the sea ice surface height or freeboard. For instance,
de Rijke-Thomas et al. (2023) recently showed that the portion of the Ku-band radar signal that is reflected at the snow-ice
interface depends closely on the altitude of the altimeter, as well as the roughness and slope distribution of the target sea ice.



The coherent radar reflection from the snow-ice interface (rather than incoherent surface or volume scattering) becomes in-

creasingly dominant as the observation altitude increases or the surface slope distribution narrows. Furthermore, a "retracking" algorithm must be applied to estimate the average elevation of the target surface from the leading edge of the waveform (Quartly et al., 2019). For sea ice altimetry, the so-called radar freeboard is then determined from the difference in retracked heights between sea ice floes and local sea surface reference samples at leads (Ricker et al., 2014). The retracking algorithm applied to floes and leads is based on a set of assumptions for their characteristic scattering properties. A bias in the derived sea ice

freeboard, versus some reference validation dataset, may therefore come from a geophysical source (e.g., pulse attenuation in the snow volume) or from the interpretation of the measurement (e.g., invalid assumptions of the retracking algorithm) (Landy et al., 2020). It can be impossible to separate these sources of error without auxiliary information.

In spite of these challenges, there have been several attempts to produce pan-Arctic snow depth estimates from dual-

frequency altimetry. Guerreiro et al. (2016) developed a methodology with CryoSat-2 observations processed in pseudo-LRM (low-resolution mode) to match the LRM AltiKa observations, and calculated snow depth at satellite crossovers <3 days apart assuming 0 and 100% snow penetration at Ka- and Ku-band, respectively. This Altimetric Snow Depth (ASD) "KuKa" processing chain was later updated to compute snow depth composites from monthly gridded CryoSat-2 and AltiKa freeboards (Garnier et al., 2021), with a comparison to snow depth estimates from Operation Ice Bridge (OIB) taken between 2014 and

2018 indicating an $R^2$ score of 0.44. Alternatively, Lawrence et al. (2018) calibrated CryoSat-2 SAR & SARIn mode and AltiKa LRM observations with coincident observations of derived radar freeboard and laser freeboard from combined airborne lidar and snow-radar spring OIB data to align the satellite freeboards from each altimeter down and up to the snow-ice and air-snow interfaces, respectively (Lawrence et al., 2018). A similar comparison to OIB data produced an $R^2$ score of 0.38. The offsets to OIB were attributed to variable radar penetration rates into snow (Armitage and Ridout, 2015), surface roughness,

and sampling differences between sensors. Calibrated freeboards averaged to monthly grids are differenced to produce KuKa snow depth estimates, after accounting for the delayed Ku-band wave propagation speed in snow. The DuST (dual-altimeter snow thickness) methodology has more recently been applied to produce pan-Arctic "KuLa" snow depth estimates from the difference between calibrated CryoSat-2 observations and uncalibrated ICESat-2 observations as part of the ESA *Polar+ Snow on Sea Ice* Project. Kwok et al. (2020) and Kacimi and Kwok (2022) have also estimated pan-Arctic KuLa snow depths from

the difference between uncalibrated CryoSat-2 and ICESat-2 observations, showing thinner depth estimates than the long-term snow depth climatology described in Kwok and Cunningham (2015). KuLa snow depth products have shown evidence for 10+ cm snow accumulation over winter and spring, matching expectations from reanalysis precipitation (Kwok et al., 2020), whereas KuKa products exhibit a much lower rate of seasonal accumulation (Lawrence et al., 2018; Garnier et al., 2021).

Here, we exploit the overlap in Arctic observations by the CryoSat-2, AltiKa, and ICESat-2 missions, at latitudes up to 81.5°N after October 2018, to examine how each sensor detects the freeboard of the snow, sea ice, or something in between. We use a waveform modelling method to simulate the theoretical radar return from rough snow and ice surfaces at Ku and Ka bands, then use these simulations to inform the retracking of surface height from CryoSat-2 and AltiKa, respectively. Monthly



winter freeboards from CryoSat-2, AltiKa, and ICESat-2 are intercompared and used to derive two different estimates of the
snow depth, which are then evaluated against independent snow depth data. Satellite laser/radar surface roughness estimates
and radar backscattering coefficients, together with airborne estimates for the sea ice and snow freeboard, are used to investigate
the wave scattering mechanisms behind differences in sensors. Finally, we consider lessons learned from these multi-frequency
observations in advance of the CRISTAL mission launch.

## 2   Data

### 2.1   CryoSat-2 SIRAL Observations

The SAR Interferometric Radar Altimeter (SIRAL) on board CryoSat-2 combines a pulse-limited Ku-band radar altimeter
with synthetic aperture and interferometric signal processing. The footprint of the sensor is therefore pulse-Doppler limited
∼300 m along the track and pulse limited to ∼1700 m across the track of the beam, with observations at ∼300 m intervals
(Wingham et al., 2006) available up to a latitudinal limit of 88°N. Here, we use the Level-1B (L1B) SAR and SARIn mode
Baseline E observations, obtained from the open ESA dissemination server at ftp://science-pds.cryosat.esa.int/, from October
2018 to April 2023, limited by the time period of ICESat-2 data (see Section 2.3) and covering the region north of 50°N.
The L1B waveform observations from the larger 240-m range window in SARIn mode are truncated to match the 60-m range
window of the SAR mode, and observations from the two modes are then treated identically. It is important to note that the
145 range resolution of the altimeter is 0.47 m; however, the waveforms in the ESA L1B product are sampled at 0.23 m after zero
padding is applied prior to the range FFT (Smith and Scharroo, 2014) to prevent aliased sampling of specular radar returns.
This should lead to improved waveform fitting of lead echoes (see Section 3.3).

### 2.2   AltiKa SARAL Observations

The AltiKa on board Satellite with ARgos and ALtika (SARAL) is a Ka-band radar altimeter, with a pulse-limited footprint of
150 ∼1400 m and beam-limited footprint of ∼8 km, producing observations at 170 m intervals (Verron et al., 2015). It is primarily
a sea level monitoring mission, with a latitudinal limit of 81.5°N. Here, we use the L1B LRM mode Geophysical Data Record
(sgdr_f) observations available from the AVISO Altimetry dissemination server at ftp-access.aviso.altimetry.fr, from October
2018 to April 2023 and covering the Arctic region north of 50°N. The range resolution of the altimeter is 0.30 m.

### 2.3   ICESat-2 ATLAS Observations

The Advanced Topographic Laser Altimeter System (ATLAS) on board ICESat-2 operates with a split-beam configuration of
three 532 nm laser beam pairs, each including a strong and a weak beam. The beam pairs are spaced ∼3 km across the track
of the sensor, with 90 m across-track and 2.5 km along-track separation between each strong and weak beam (Markus et al.,
2017). The high pulse-repetition frequency produces laser pulses every ∼70 cm along track with a footprint diameter of ∼11



m. Here, we use the Level-3B ATL20 Version 4 Daily and Monthly Gridded Sea Ice Freeboard product available from NSIDC at Petty et al. (2023b), from October 2018 to April 2023. The laser freeboard is produced by aggregating 150 photons along a beam, determining the sea ice height from the photon distribution of this segment, and finding the height difference to a reference sea surface height determined from the photon distribution at local lead points (within 10 km along-track sections for each beam) (Kwok et al., 2019b). For ATL20, the along-track freeboards are aggregated at daily and then monthly timescales onto a 25-km x 25-km Polar Stereographic North projection grid across the Arctic which we resample onto an EASE2 grid using nearest neighbour interpolation. Only the three strong beams are used to produce ATL20. To obtain an estimate for the surface roughness that is compatible with ATL20 and on a similar scale to the CryoSat-2 observations, we use the Level-3A ATL07 Version 6 Sea Ice Height product available from NSIDC at Kwok et al. (2023). Roughness is estimated from the standard deviation of surface heights from the central strong beam within 25-km along-track sections. Samples are removed if the section contains <100 heights, then binned onto the same 25-km EASE2 grid.

Furthermore, we use the ICESat-2 Arctic Sea Ice Surface Topography data Version 2.1 from the University of Maryland-Ridge Detection Algorithm (Duncan and Farrell, 2022), available at https://zenodo.org/records/7129192. We use the snow surface roughness parameter, obtained from a ridge detection scheme applied to ATL03 photon heights, for April 2019, to support the simulation of CRISTAL waveforms (Section 5.4).

## 2.4 Reference Observations

To validate our satellite snow depth estimates and investigate the radar backscatter mechanisms potentially introducing biases into freeboard observations, we compare the satellite measurements to several airborne and *in situ* datasets. This includes a set of reference observations prepared and gridded by AWI for the ESA Polar+ Snow on Sea Ice Project: (i) Snow depth from airborne radar data available in the AWI IceBird Winter 2019 campaign dataset (Jutila et al., 2021), accessible from https://doi.pangaea.de/10.1594/PANGAEA.933912, for April 2019. These data were processed with the *PySnowRadar* package (https://doi.org/10.5281/zenodo.4071947)) based on the "peakiness" method described in Jutila et al. (2022). (ii) Snow depth collected manually along the MOSAiC campaign transects (Itkin et al., 2023), accessible from https://doi.org/10.1594/PANGAEA.937781, for October 2019 to April 2020. (iii) Snow depth processed from airborne radar data collected by The Center for Remote Sensing and Integrated Systems (CReSIS) at the University of Kansas with the same *PySnowRadar* parameters as the IceBird data, accessible from data.cresis.ku.edu/#SR, for all five of the OIB flight campaigns available in April 2019. The reference observations are gridded onto a monthly 25-km EASE2 (Brodzik et al., 2012) projection to match the satellite data.

Furthermore, we use the laser scanner-derived snow (or total) freeboard observations from the April 2019 OIB and AWI airborne campaigns, along with the airborne snow depth estimates, to estimate the expected Ku-band freeboard – under the assumption of total radar penetration – following the approach of Lawrence et al. (2018). The sea ice freeboard is calculated by subtracting the airborne radar snow depths from the laser scanner snow freeboards, then converted to the expected Ku-band



radar freeboard by accounting for the delayed radar wave propagation through snow (assuming a snow density of 350 kg m$^{-3}$). The laser scanner snow freeboard observations represent the expected Ka-band radar freeboard, under the assumption of zero

radar penetration.

## 3   Methods

### 3.1   Basis of the approach

The basis for the waveform modelling approach (The Lognormal Altimeter Retracker Model; LARM) is to fit a physical model for the backscattered radar altimeter echo to observed altimeter waveforms and obtain, through the model inversion (Landy

et al., 2020), estimates for the retracked snow-ice interface elevation (CryoSat-2), air-snow interface elevation (AltiKa), or sea surface height at leads (both sensors). A mean sea surface (MSS) model, here the DTU21 solution (Baltazar Andersen et al., 2023), is subtracted from the retracked heights to calculate sea ice height and sea surface height anomalies (SSHA). The radar freeboard is calculated from the height difference between the ice floe elevation anomalies and an estimate for the SSHA interpolated between tie-points from nearby leads along the orbital track of the satellite. Radar freeboards from a given month are

then binned and averaged on a 25-km x 25-km EASE2 grid that covers the entire Arctic and finally corrected for the delayed travel speed of the CryoSat-2 Ku-band radar wave through snow (Mallett, 2024) to produce estimates for the pan-Arctic snow depth, following e.g. Lawrence et al. (2018). These steps are explained in more detail below.

The fundamental assumption of the method is that forward model solutions for the CryoSat-2 Ku-band SAR echo or AltiKa

Ka-band LRM echo adequately represent the height of the snow-ice or air-snow interface, respectively, based on the physical mechanisms scattering or reflecting the radar waves at each frequency. Here we take the simple but necessary approach to model **both Ku-band and Ka-band radar waveforms as if the backscatter is returned from a single backscattering surface**, assumed to represent the snow-ice and air-snow interfaces, respectively. Several studies have called this assumption into question, as described in the introduction. For instance, snow volume scattering, attenuation by brine-wetted snow, new

snowfall, and variations in radar-scale surface roughness may impact the height of the maximum radar backscattering intensity at each frequency. In those cases our fundamental assumption of a single scattering surface is invalidated and/or the model inadequately reproduces the true scattering response of the radar wave, we will obtain a bias in the retracked elevation.

The alternative option would be to forward model the full combination of possible surface and snow volume scattering mech-

anisms, e.g. see Section 5.4. However, any influential model parameter that cannot be constrained by additional data (e.g., snow depth, density, grain size, dielectric properties, etc.) would need to be a free parameter in the model inversion. In this scenario, the inversion of a larger number of free parameters based on a single waveform would consequently be very uncertain with high degeneracy between certain model parameters, for example, the surface roughness and snow depth (de Rijke-Thomas, 2019), so this is currently not considered a viable option.




## 3.2 Radar waveform modelling

In the case of a single backscattering surface the radar altimeter echo model is parameterized by four terms: $A$, the scaled wave-form amplitude, $t_0$, the tracking point of the mean radar scattering surface (or 'epoch'), $\sigma$, the surface topography root-mean square height, and $s_{rms}$, the mm-cm 'radar-scale' roughness. This is assuming that variations in the radar antenna parameters
(e.g., satellite altitude, off-nadir pointing angle) have a negligible impact on the shape of the sea ice waveform return and can be ignored.

We use the Facet-Based Echo Model (*FBEM*) which simulates the radar altimeter echo as the integral of the power backscat-tered from a tetrahedral mesh representing the sea ice surface topography (Landy et al., 2019). The FBEM is available as an
open-source *MATLAB* code from https://github.com/jclandy/FBEM. The scattering mechanisms are characterized for each facet of the mesh. Backscatter from the air-snow or snow-ice interface is obtained from the sum of the scattered and reflected components, simulated with the Integral Equation Model (*IEM*; (Fung and Chen, 2004)) and Kirchoff physical optics approx-imation (Fung and Eom, 1983), respectively. The balance between scattering and reflection depends on the radar frequency and radar-scale roughness of the surface (Landy et al., 2020). A waveform simulation uses a single radar-scale roughness for
every facet; however, the backscatter from each facet still varies nonlinearly as a function of the facet's local slope angle – being larger for smaller slope angles that face the radar (Landy et al., 2020; Nandan et al., 2023). This means level ice facets contribute disproportionally to the total backscattered echo, compared to sloped facets (i.e., ridges), and reduce the height of maximum backscattered radar intensity compared to the true mean floe surface height (de Rijke-Thomas et al., 2023). This is analogous to the radar altimetry sea state bias over the ocean (Tran et al., 2010).


The large-scale topographies of the air-snow or snow-ice interfaces are characterized by a Lognormal probability density function of the height distribution, following Landy et al. (2020). A single lookup table of Ku-band SAR altimeter echoes is simulated from FBEM for sea ice and lead surfaces with lognormal height PDF, $\sigma$ ranging from 0 to 1 m, and $s_{rms}$ ranging from 0 to 6 mm, as described in Landy et al. (2020). Radar antenna parameters are characterized for the CryoSat-2 SIRAL
instrument (Landy et al., 2019). Examples for the modelled SAR echoes with fixed $s_{rms}$ of 2 mm but varying $\sigma$ are shown in Figure 1a. The parameter $s_{rms}$ controls the magnitude of the radar backscatter from the surface and the incidence angle dependence of the backscattering coefficient, with a smaller value producing a higher power and peakier waveform return. The parameter $\sigma$ controls the roughness of the large-scale sea ice topography and principally affects the width of the waveform leading edge. When $s_{rms} = 2$ mm and $\sigma = 0$ (orange waveform) the echo represents the approximate reflection of the trans-
mitted pulse. Such peaky waveforms are characteristically produced when the radar signal is reflected from leads in sea ice. The radar tracking point $t_0$ defines the mean elevation of the surface height distribution within the radar footprint. Zero time on Figure 1 represents $t_0$ and is crossed at a different amplitude of the waveform leading edge power depending principally on $\sigma$. This implies that the relative retracking amplitude should decrease as the large-scale roughness of the snow-ice interface



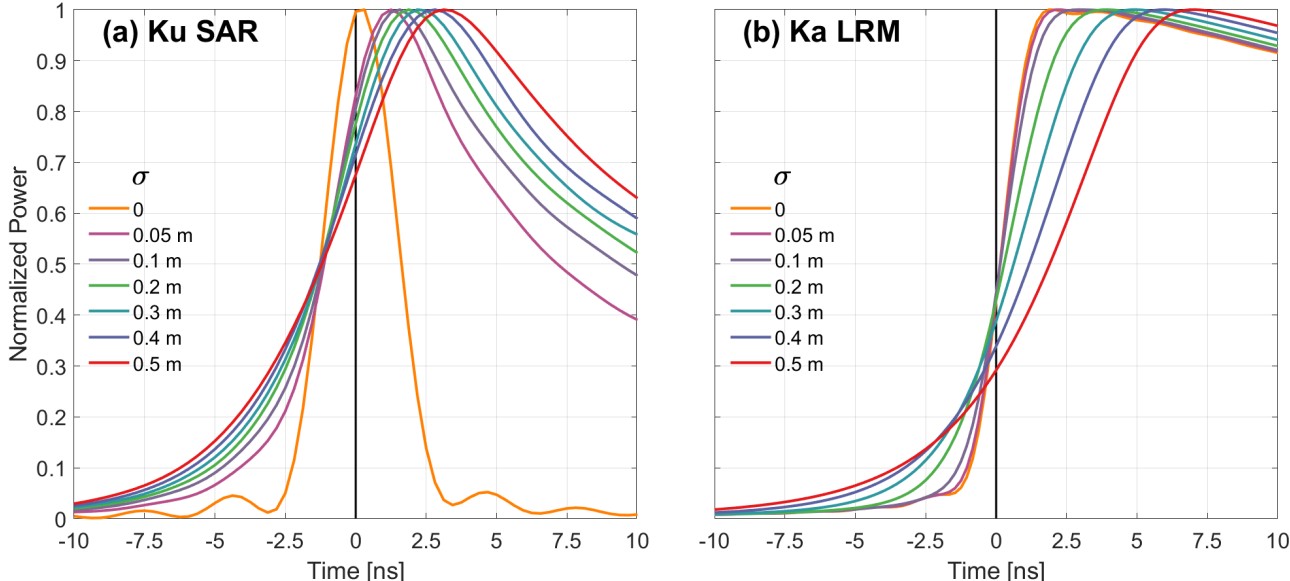

**Figure 1.** Radar echo simulations for a single backscattering interface (snow or sea ice), with a Lognormal roughness height distribution and different roughness standard deviations $\sigma$, in (a) Ku-band SAR-mode with radar sensing parameters from CryoSat-2 SIRAL and (b) Ka-band LRM-mode with radar sensing parameters from AltiKa SARAL. For these examples $s_{rms}$ is held at a fixed 2 mm. Zero time represents the radar tracking point $t_0$.

increases, from around 95 8% for $\sigma$=0 to 67% for $\sigma$=0.5 m (i.e., tracing down the $t$=0 line on Figure 1a).


A single lookup table of Ka-band LRM altimeter echoes is also simulated from FBEM for sea ice and lead surfaces with Lognormal height PDF, $\sigma$ ranging from 0 to 1 m, and $s_{rms}$ ranging from 0 to 6 mm. Radar antenna parameters are characterized for the AltiKa SARAL instrument, based on https://directory.eoportal.org/web/eoportal/satellite-missions/s/saral. Examples for the modelled LRM echoes with fixed $s_{rms}$ of 2 mm but varying $\sigma$ are shown in Figure 1b. The roughness of the large-scale

sea ice topography $\sigma$ appears to impact the width of the leading edge more in Ka-band LRM-mode than in Ku-band SAR-mode (e.g., Guerreiro et al., 2016; Fredensborg Hansen et al., 2021). The radar tracking point $t_0$ is also again crossed at a different amplitude of the waveform leading edge power depending on $\sigma$ (i.e., tracing down the $t$=0 line on Figure 1b). This implies that the relative retracking amplitude for LRM waveforms should also decrease as the large-scale roughness of the air-snow interface increases, from around 50% for $\sigma$=0 to 28% for $\sigma$=0.5 m.




### 3.3 Waveform fitting and freeboard derivation

We use a least-squares fitting procedure to optimize the functional form of the modelled sea ice echo to observed CryoSat-2 or AltiKa waveforms. The fitting routine is based on the bounded trust region reflective algorithm (implemented through the MAT-LAB function *lsqnonlin*) to minimize the difference between the model fit and each observed power waveform, as described in Landy et al. (2020). A filtering routine is applied to exclude samples at major secondary peaks, on the waveform trailing edge, from the model fit (Figure 2). At the end of the fitting procedure, we obtain optimal estimates for the four free parameters: A, $t_0$, $\sigma$, and $s_{rms}$. Examples for the best-fitting modelled echoes to sea ice floe surfaces and leads, for both CryoSat-2 and AltiKa waveforms, are shown in Figure 2 along with their associated parameters. It is evident that the large-scale topography $\sigma$ and small-scale roughness $s_{rms}$ are both larger for the floe surfaces than for leads. It is also notable that the zero padding applied to CryoSat-2 SAR waveforms doubles the range sampling, without adding any new information (Smith and Scharroo, 2014), but provides an improved constraint on the elevation of specular leads than can be obtained from the specular AltiKa LRM waveforms at their native bandwidth-limited range resolution (Figure 2b and d).

The satellite range is obtained from half the retracked two-way travel time to the surface multiplied by the speed of light. The surface height relative to the WGS84 ellipsoid is then obtained from the satellite altitude minus the range and is subsequently corrected for atmospheric effects (dry and wet troposphere, ionospheric delay, inverse barometer correction) and geophysical effects of the ocean (ocean tide, loading tide, pole tide) using corrections provided in the data products. The corrections can therefore be different between sensors, with negligible impact on the results (Ricker et al., 2016). A conservative low-pass filter is applied to the along-track height profile to remove residual sea surface topography (with a horizontal wavelength greater than 200 km) that is not removed by the geophysical corrections.

Waveforms are separated into three classes (sea ice floes, leads, and ocean) with a simple thresholding technique based on several waveform shape parameters. No class is used for "ambiguous" waveforms. For CryoSat-2, the parameter thresholds are based on results from Müller et al. (2023) who showed that previous classification schemes for CryoSat-2 generally assigned observations over thin sea ice to the ambiguous class, potentially biasing the derived sea ice freeboard high by omitting them. With revised classes the waveforms previously classed as ambiguous are now generally classed as sea ice. Observations are classed as leads where the radar backscattering coefficient ($\sigma^0$) is >23 dB in SAR mode and >24 dB in SARIn mode, the pulse peakiness ($PP$) is >0.258 in SAR mode and >0.254 in SARIn mode, and the leading-edge width ($LEW$) is <4.69 ns in SAR mode and <6.56 ns in SARIn mode (Hendricks et al., 2021). Observations are classed as ocean where $\sigma^0$<2.5 dB, the stack standard deviation ($SSD$) >55, or the sea ice concentration (from the OSISAF product *OSI-401-d*) <15%. All remaining valid observations are classified as sea ice. For AltiKa, the parameter thresholds are based on Armitage and Ridout (2015) and Zakharova et al. (2015). Observations are classified as leads where $\sigma^0$>15 dB, the $PP$>0.156, and the $LEW$<4.58 ns. Observations are classed as ocean where $\sigma^0$<2.5 dB or the sea ice concentration <15%. All remaining valid observations are





**Figure 2.** Best fitting model echoes to observed radar waveforms from (a,b) CryoSat-2 SAR-mode and (c,d) AltiKa LRM-mode. (a) and (c) show model fits to characteristic diffuse-type waveforms returned from a rough sea ice and/or snow surface. (b) and (d) show model fits to characteristic specular-type waveforms returned from smooth ocean lead surfaces. The dashed lines mark the 'epoch' or radar retracking point, i.e. the mean level of the scattering surface, with respect to the echo maxima. Hollow samples at major secondary peaks are discarded from the waveforms during fitting.



classified as sea ice.


The SSHA is obtained at ice-covered locations from a linear interpolation between lead elevations, along the orbit of the satellite, as described in Landy et al. (2020). All sea ice observations >300 km from their nearest lead are discarded. Along-track radar freeboards are calculated from the elevation difference between sea ice floes and the interpolated SSHA. An estimate for the SSHA uncertainty at a lead location is made from the standard deviation of lead elevations within a 50-km along-track
window around the lead. At ice floe locations, the SSHA uncertainty is estimated from the uncertainty at proximal leads, scaled by the inverse of the squared distance to the nearest lead, up to a maximum of 10 cm at 300 km.

### 3.4   Snow depth estimation and intercomparison

The radar freeboards for both CryoSat-2 and AltiKa are gridded, at monthly intervals, onto the same 25-km x 25-km EASE2
projection Arctic grid using a binned weighted mean algorithm, with each freeboard observation weighted inversely by its absolute uncertainty. The SSHA uncertainty is gridded with the same method. The range error on a single 20 Hz CryoSat-2 observation is estimated to be 10 cm for SAR mode and 14 cm for SARIn mode (Wingham et al., 2006) and the range error on a single 40 Hz AltiKa LRM mode observation is estimated to be 5 cm (Dettmering et al., 2015). This range error is considered fully random, incorporating speckle and retracking error from the waveform fitting process; however, there may be other sys-
tematic errors, such as retracker bias, that are not known and thus not included. For a conservative estimate of the total radar freeboard uncertainty, we therefore sum the range error (multiplied by $1/\sqrt{N}$ observations in a grid cell) and the mean gridded SSHA uncertainty in quadrature. After averaging to a 25-km scale, the precision on the gridded radar freeboards is around 2-3 cm (Fredensborg Hansen et al., 2024).

The ICESat-2 ATL20 monthly snow freeboard data are reprojected onto our EASE2 grid. We then estimate the snow depth in two ways: (i) from the height difference between CryoSat-2 and AltiKa, "KuKa", and (ii) from the height difference between CryoSat-2 and ICESat-2, "KuLa". We assume that the CryoSat-2 radar freeboards represent the elevation of the snow-ice interface (see Section 5.4 below), but this is not yet corrected for the delayed Ku-band wave propagation velocity through the snowpack. To derive estimates for the snow depth from the height difference in gridded freeboards, we therefore multiply by
the height difference corresponding to the ratio of radar wave velocities in snow and free space (Lawrence et al., 2018). The snow density is estimated to vary seasonally, between 266 and 329 kg m$^{-3}$, from October to April, based on Mallett (2024). Snow depth uncertainty is derived from the radar freeboard uncertainties and their covariance, for KuKa, and from the radar and laser freeboard uncertainties and their covariance, for KuLa, following the method of Lawrence et al. (2018). (Note that we do not calibrate the radar freeboards, as in Lawrence et al. (2018), so uncertainties on the calibrations are also omitted from
the uncertainty calculation).





# 4 Results

We first analyze the basin-wide patterns and distributions of the CryoSat-2 (CS2) and AltiKa (AK) radar freeboards, and ICESat-2 (IS2) laser freeboards, in two selected months: December 2018, in the early stages of the snow accumulation season
but with sea ice close to its maximum extent, and April 2019, towards the end of the same snow accumulation season. We then analyze the differences between the KuKa and KuLa versions of the snow depth, including the seasonal differences in estimated snow accumulation rate. Finally, we intercompare the derived snow depths with airborne and *in situ* snow depth measurements collected between 2019 and 2020.

## 4.1 Radar and laser freeboard intercomparison

The CryoSat-2 radar freeboard observations in December and April are thinner than both the AltiKa radar and ICESat-2 snow freeboards. All three datasets show relatively thicker freeboards over the MYI than over the FYI zone (Figures 3a-c and 5a-c). The mean of the CS2 freeboard distribution is 3.7 and 5.3 cm, in December and April respectively, compared to 21 and 26 cm for AK and 18 and 30 cm for IS2 (Figures 4a and 6a). Around half of the December CS2 freeboards south of 81.5°N are
thinner than 3 cm, representing new ice forming recently in the marginal ice zone (MIZ). This threshold is approximately the lower freeboard detection limit of CryoSat-2 when using a physically-based retracker (see Figure 7 in Landy et al. (2020) and Fredensborg Hansen et al. (2024)). Thicker AK and IS2 freeboards in the MIZ represent snow accumulating on newly-forming sea ice (see 5.3).

The AK and IS2 freeboards display many similarities in their regional variability, for example thicker freeboards in the northern region of the Canadian Arctic Archipelago (CAA), in the Beaufort Sea, and in the Fram Strait. The transition in freeboard across the MYI tongue circulating into the Beaufort Sea, in particular, is captured by both AK and IS2 in December and April. It is evident, however, that the AK radar freeboards show more local spatial variability than the smoother patterns of the CS2 and IS2 freeboards, which may be attributable to higher uncertainty in the AltiKa LRM freeboard observations. All three
sensors exhibit freeboard patterns in December that closely match the ice type zones mapped in the OSI-403-d product, with sharp gradients in freeboard across the MYI edge (Figure 3a-c). It is notable that by April the clear separation by ice type is not preserved in the CS2 freeboards and, while it is more obvious in the AK and IS2 freeboards, it is not in the Atlantic Sector where the ice is more dynamic and seasonal ice thickening/snow accumulation might depend less on the ice type (Figure 5a-c; see Section 5.3).


The CS2 radar freeboard distributions are narrower than the others with ~15% of grid cells having near-zero or slightly negative freeboards, reflecting thinner sea ice floes loaded by surface snow (Figures 4a and 6a). MYI freeboards are evident in the tails of the CS2 distributions thicker than ~10 cm. The AK and IS2 freeboard distributions are both unimodal in December, but the IS2 distribution alone becomes bimodal by April, reflecting thickened IS2 freeboards in the Chukchi and East



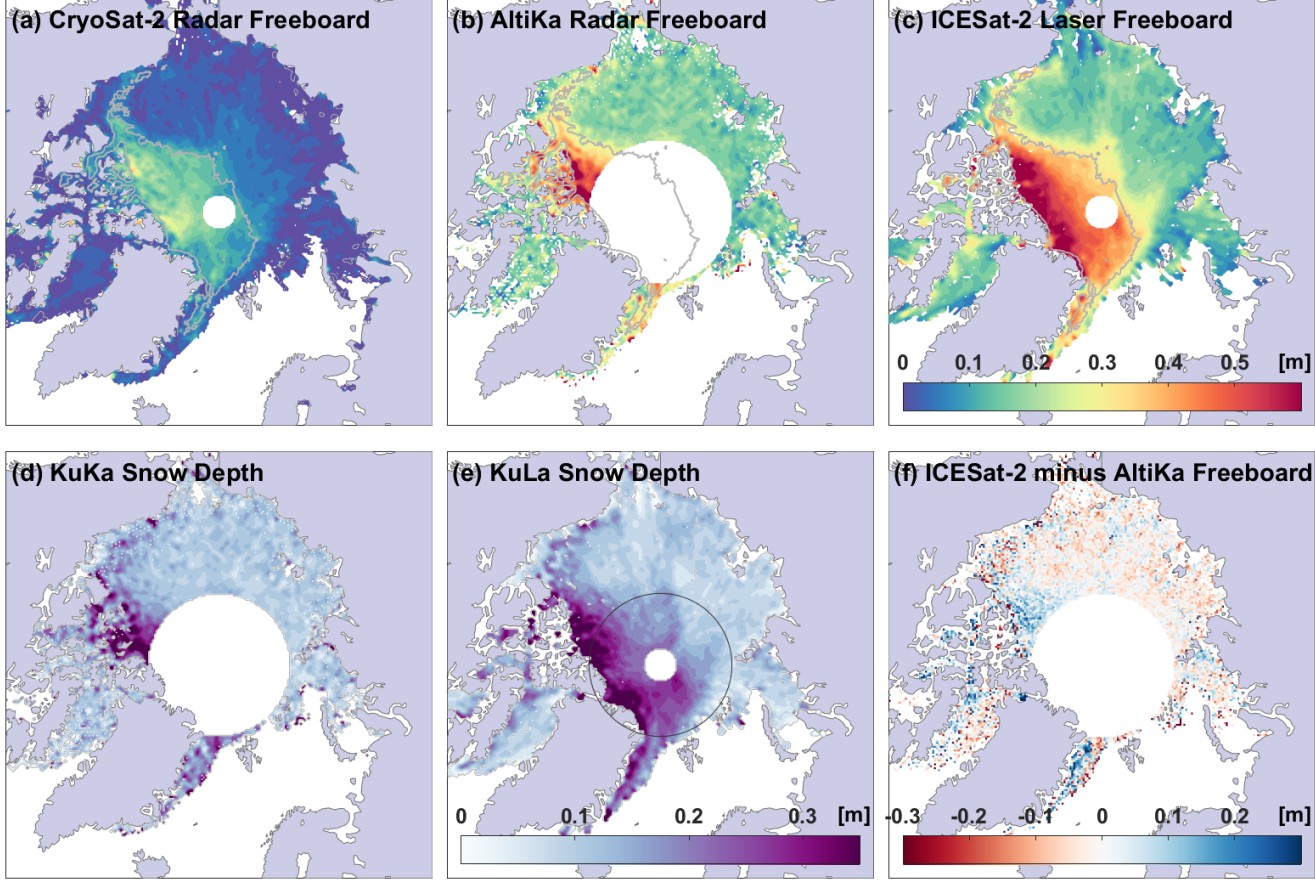

**Figure 3.** Example of the CryoSat-2 (a) and AltiKa (b) radar freeboards obtained from physical-model waveform fitting in December 2018, with comparison to ICESat-2 (c) laser freeboards. Two estimates for the snow depth are obtained from a simple difference between KuKa (CryoSat-2 and AltiKa) freeboards (d) and KuLa (CryoSat-2 and ICESat-2) freeboards (e) corrected for the delayed Ku-band wave speed through the snow volume. A map of the difference between ICESat-2 and AltiKa freeboards is shown in (f). The grey lines on (a)-(c) show the boundary of the MYI zone based on the monthly-mean sea ice type from OSI SAF (OSI-403-d).

Siberian Seas that are not observed in the AK data and only weakly observed in the CS2 data. Generally, the freeboard patterns from CS2 and IS2 are quite similar, albeit with different magnitudes. The thinnest CS2 freeboards (<5 cm) in the MIZ typically coincide with the thinnest IS2 freeboards (<10 cm) (Figures 3a and c and 5a and c), where snow depth is expected to be thinnest over newly-formed sea ice (see Section 5.3). Clear exceptions are in the Chukchi Sea, where IS2 freeboards thicken more than any other region between Dec 2018 and Apr 2019 (Kwok et al., 2020) and in the Baffin Bay where the east-

west gradient in IS2 freeboards is not so evident in the CS2 freeboards (Glissenaar et al., 2021). The IS2 freeboards also show a broader distribution of thickness (std dev 15.3 cm), across the entire Arctic, in April, than the CS2 freeboards (std dev 6.7 cm).



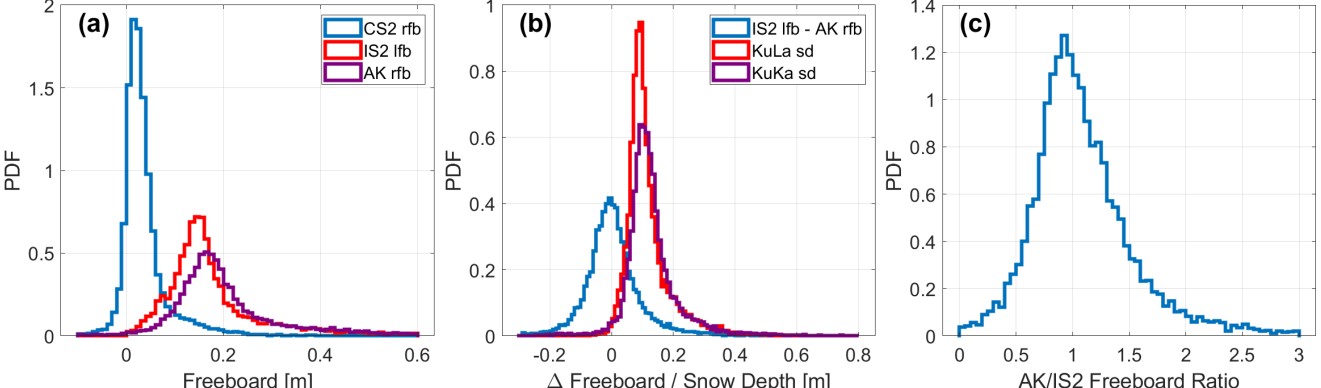

**Figure 4.** Comparison of the gridded radar freeboards (rfb) obtained from physical model waveform fitting applied to CryoSat-2 and AltiKa to laser freeboards (lfb) obtained from ICESat-2 in December 2018. (a) Freeboard distributions. (b) Derived snow depth (sd) distributions and the distribution of gridded differences between AltiKa and ICESat-2. (c) The AltiKa freeboards plotted as a ratio of the ICESat-2 freeboards. (Note that distributions only cover the coinciding region of observations between the three sensors south of 81.5°N).

## 4.2 Snow depth intercomparison

The KuKa and KuLa snow depths are shown in Figures 3d-e and 5d-e. There are some clear similarities between the two prod-
ucts, including thicker derived snow depths over MYI north of Canada and in Fram Strait, than over FYI in the surrounding Arctic seas. The locations of thicker snow depths in the northern CAA and in the Beaufort Sea are very similar. However, the KuLa maps show more regional variability in snow depth than the KuKa maps, including within the first-year ice/MIZ regions, where snow depth is sensitive to freeze-up timing, strong snowfall events, and wind compaction/snow loss events (Webster et al., 2018). For instance, there is no clear gradient in snow depth at the Barents Sea or Baffin Bay ice edge, in the December
KuKa product (Figure 3d). The KuKa snow depths exhibit more local 10s km-scale variability, reflecting the variability of the AK radar freeboards, whereas the KuLa snow depths exhibit smooth regional gradients in snow depth over scales of 1000s km. The pan-Arctic (up to 81.5°N) snow depth distributions are relatively narrow and similar in December (Figure 4) but diverge in April, with KuKa snow depths showing a mode at 15 cm and KuLa snow depths showing a broader peak with mode at 19 cm but small secondary peak of thinner depths at ∼10 cm.


The height differences between AK and IS2 freeboards show some regional patterns. In December, IS2 is ∼3-4 cm thicker than AK over the MYI region north of Canada, but generally similar to AK over FYI, albeit with local grid-cell scale differences (Figure 3f). In April, AK freeboards generally underestimate those from IS2, by 3-15 cm, except around the ice edge (Figure 5f). Figures 4c and 6c show that the distributions of the grid-cell freeboard ratio between AK and IS2 have a mode around 1 in
December and around 0.85 in April. The shift in freeboard ratio for April is particularly caused by IS2 observing much thicker freeboards than AK in the Chukchi and East Siberian Seas (Figure 5f). The distributions are approximately Gaussian around





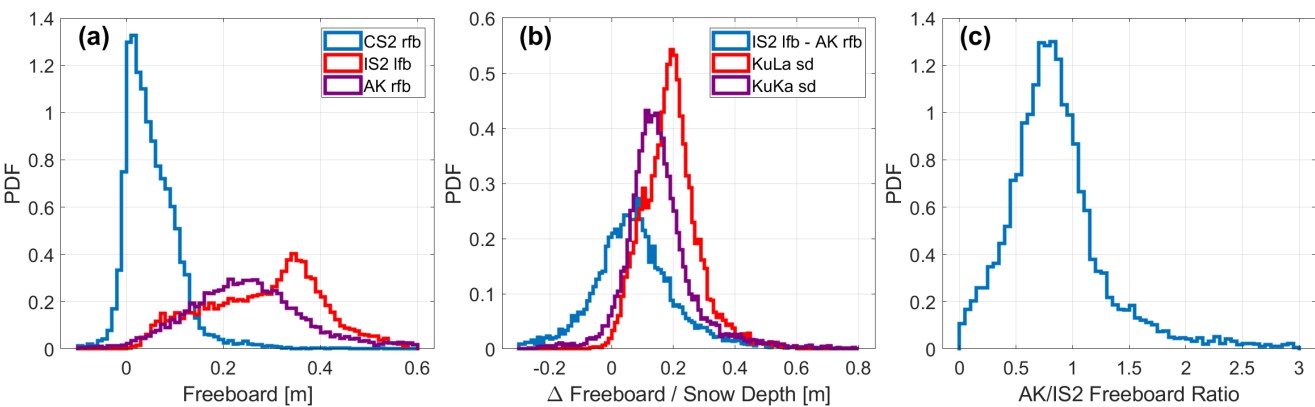

**Figure 5.** Same as Figure 3, but for April 2019

**Figure 6.** Same as Figure 4, but for April 2019





these modal values, except from a tail of grid cells with higher ratio, where AK freeboards are 50-100% thicker than those measured by IS2.

### 4.3 Seasonal snow accumulation

The time series of KuKa and KuLa mean snow depth and variability, across the five accumulation seasons, exhibit some clear differences (Figure 7). Both time series show increasing snow depths through winter, with the exception of October to November when large areas of newly-forming sea ice with lower accumulated snow reduce the basin-wide mean. The mean and standard deviation of the snow depth are also both higher over MYI than FYI, for both products. However, the basin-wide standard deviation in the snow depth is around 8-13 cm for KuKa compared to 6-10 cm for KuLa. The rates of estimated snow accumulation are also quite different between the products, with a mean seasonal increase of 0.5 cm/month for KuKa compared to 1.7 cm/month for KuLa. The rate of snow accumulation for the KuKa product is implausibly low. For the KuLa product, the pan-Arctic mean rate of accumulation including observations up to 88°N, is 1.6 cm/month (Figure A1). Relatively low rates of snow accumulation have been observed previously for CryoSat-2-AltiKa snow depth products (Lawrence et al., 2018; Garnier et al., 2021).

The KuLa snow depths, which offer near basin-wide coverage, grow from around 8 to 18 cm over first-year ice and 17 to 26 cm over multi-year ice across the snow accumulation season (Figure 7b). The accumulation of snow on MYI is slower than on FYI, however, at a mean accumulation rate of around 1.6 cm/month on MYI versus around 1.9 cm/month on FYI. The interannual variability in KuLa snow depth is 0.4 and 0.6 cm, respectively, at the start and end of the snow accumulation season, across these five years. This is significantly lower than the 3-7 cm interannual variability in snow depth obtained from a Lagrangian snow accumulation scheme (SnowModel-LG) and would make only a small contribution to the estimated interannual variability of sea ice thickness (20-30 cm) (Mallett et al., 2021).

### 4.4 Evaluation against independent observations

The satellite snow depths are compared with *in situ* observations and airborne snow depth estimates (see Section 2.4) in Figure 8. It is important to note that KuKa observations are compared to reference data covering only $N = 50$ of our 25-km x 25-km grid cells in April 2019, with a low density of leads for some grid cells in the northern channels of the Canadian Arctic Archipelago. Our KuLa snow depths are compared to the same data plus additional reference data at higher latitudes covering a total of $N = 188$ grid cells across eight months. Each satellite product overestimates the reference snow depth by 2 cm, on average, and tends to overestimate the thinner snow depths over FYI, in particular. The KuKa product has a slope of 0.58 with respect to the reference data, meaning that it underestimates the spread of snow depths at a 25-km scale. The thinnest reference snow depths are overestimated and the thickest underestimated by ∼5 cm. For KuLa also, reference snow depths from the thinnest category of 5-10 cm, collected over FYI in the Beaufort Sea and from the MOSAiC transects in autumn, are consistently overestimated. However, the slope of the KuLa product is 0.86 with respect to the reference data meaning that it





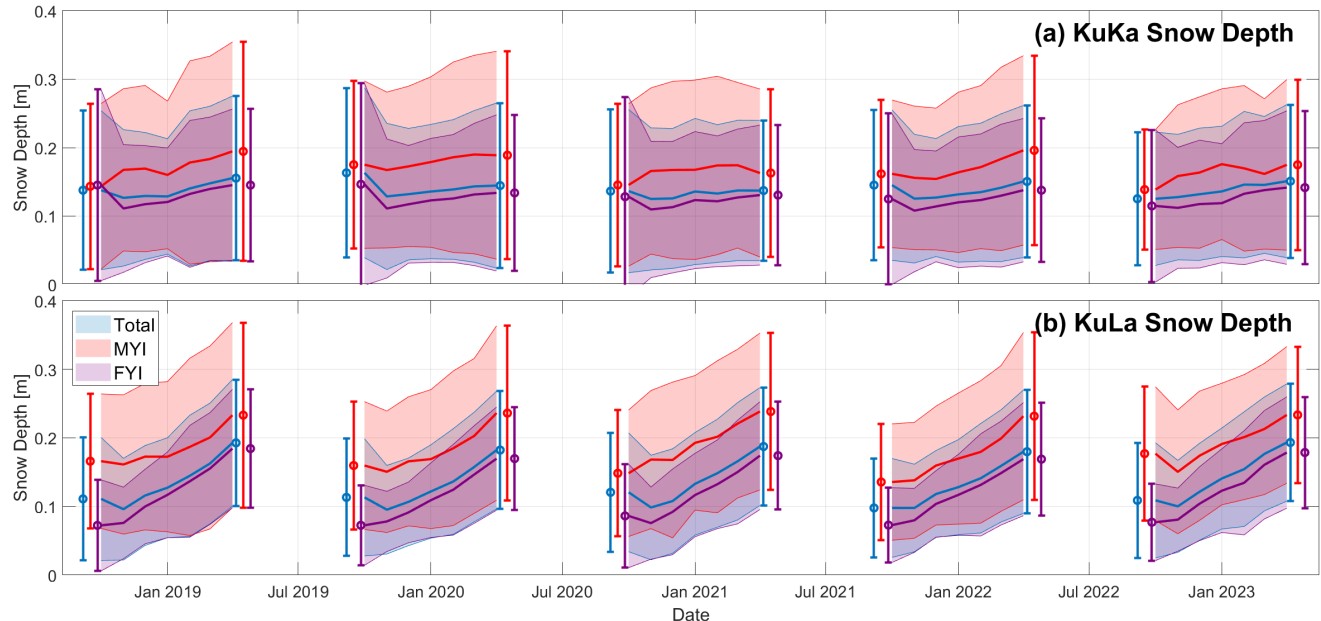

**Figure 7.** Time series for the seasonal change in snow depth obtained from (a) KuKa radar and (b) KuLa radar and laser freeboards over the 2018-2023 sea ice growth/snow accumulation seasons, for coincident data up to 81.5°N. The envelopes represent +/- one standard deviation around the mean snow depth. The points and whiskers also show mean and one standard deviation snow depths at the start and end of each observation season. A second version of this figure, including KuLa observations up to 88°N is shown in Appendix Figure A1.

generally represents the spread of the snow depths at 25-km very closely. The KuLa product reasonably matches the magnitude and variability of the reference snow depths, with a correlation of 0.79 and RMSE of 7 cm (Figure 8). For fewer grid cells, south of 81.5°N, the KuKa product has a correlation of 0.70 and RMSE of 9 cm versus the reference snow depths. When grid cells north of 81.5°N are excluded from the KuLa comparison, the correlation is 0.80 and RMSE still 7 cm; however, the bias increases to 3 cm and slope reduces to 0.72.

## 5   Discussion

It is challenging to identify the source or sources of bias between the KuKa and KuLa snow depths in any given month, and between each product and the reference snow depth data. Biases could be caused by (i) the height of maximum intensity of the Ku- or Ka-band radar backscatter not aligning with the snow-ice or air-snow interface, respectively, owing to radar penetration or surface roughness-related effects. (ii) The ICESat-2 laser penetrating into the snowpack (Studinger et al., 2024) or possible lead height retrieval errors leading to uncertainties in the laser freeboards (Kwok et al., 2019a). (iii) Our model for the radar altimeter not adequately simulating the true backscattered radar response in SAR or LRM mode. (iv) Our model assumption of a single backscattering surface not being valid, owing to strong contributions from snow volume scattering or reflections





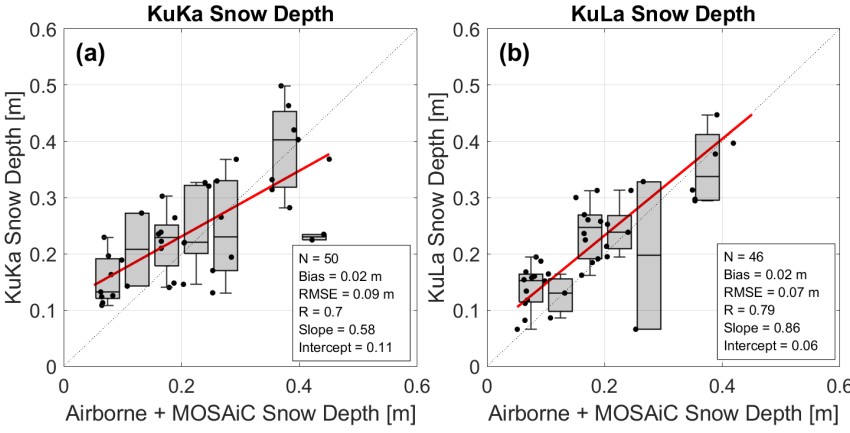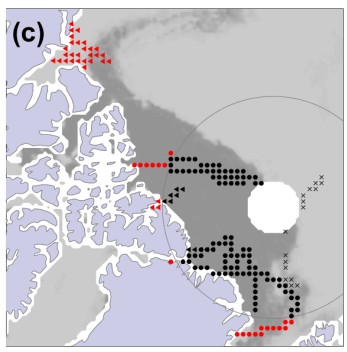

**Figure 8.** Intercomparison between snow depth estimates from (a) KuKa radar freeboards, (b) KuLa radar and laser freeboards, and independent snow depth observations. Box and whiskers are binned at 5-cm intervals. The map in (c) shows sea ice type from OSI-403-d, with MYI in dark grey and FYI in light grey, for April 2019, with the locations of data from OIB April 2019 (circles), IceBird April 2019 (triangles), and the MOSAiC Transect October 2019 - April 2020 (crosses). Red samples are compared to KuKa and KuLa satellite snow depths, whereas black samples are compared only to KuLa snow depths.

from other interfaces than the one modelled. (v) Inaccurate sample classification erroneously removing sea ice floe or including lead elevation measurements in the freeboard products (e.g., Petty et al., 2021; Fredensborg Hansen et al., 2021; Müller et al., 2023). (vi) Different orbital sampling between sensors and interpolation errors mapping observations to monthly pan-Arctic grids ((Lawrence et al., 2019). The reference data themselves are also uncertain, especially measuring the thinnest and thickest snow depths (Kwok et al., 2017). Any one or a combination of these effects could produce a mismatch between the AltiKa and ICESat-2 freeboards, that we are assuming both measure the height of the air-snow interface, or an error in derived KuKa or KuLa snow depth. Here we investigate some of these possible causes.

## 5.1 Radar waveform retracking

The radar waveform retracking algorithm can have a significant impact on the obtained CryoSat-2 or AltiKa radar freeboard (Landy et al., 2020), even though its application can be quite subjective. For instance, the choice of threshold for an empirical algorithm or assumed sea ice scattering properties for a physical model-based algorithm. Here, we assume that retracked heights from AltiKa correspond to the height of the air-snow interface, but they may in fact measure a depth well into the snowpack. We therefore estimate the AK floe retracking threshold that would be required to exactly match the AK radar freeboard to either the IS2 laser or CS2 radar freeboard and compare it to the solution derived here from LARM. This should indicate whether the retracking thresholds generated by the model echoes in LARM (Figure 1b) produce freeboards closer to the air-snow or snow-ice interface height of the sea ice, based on IS2 and CS2 freeboards, respectively. For this experiment, the AK lead waveforms in December 2018 are retracked with LARM (i.e., Figure 2d) then each floe waveform is retracked with





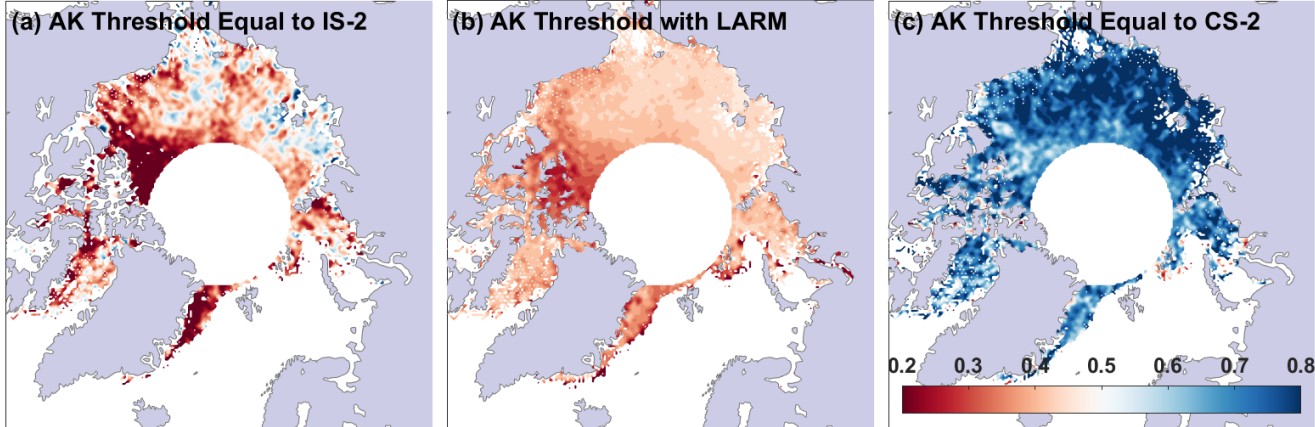

**Figure 9.** Fractional retracking thresholds of the first primary peak of the AltiKa waveform leading edge (a) required to match the AltiKa radar freeboard to the ICESat-2 laser freeboard, (b) obtained from the physical retracking method LARM, and (c) required to match the AltiKa radar freeboard to the CryoSat-2 radar freeboard, averaged within EASE2 25-km grid cells, in December 2018.

an empirical threshold method at 2.5% intervals from 5 to 95% of the first primary peak. For each 25-km grid cell in Figure 9 we find the retracking threshold that best matches the grid-cell mean AK radar freeboard to the grid-cell mean (a) IS2 laser freeboard and (c) CS2 radar freeboard.

    Previous studies have used a fixed 50% threshold for retracking sea ice floe waveforms from AltiKa (Armitage and Ridout, 465    2015; Garnier et al., 2021). The grid-cell mean AK retracking threshold that is optimized from the physical echo model with LARM is shown in Figure 9b. There is a clear pattern with thresholds around 30% in the MYI zone north of the Canadian Arctic, then a gradient towards thresholds of 45% over younger FYI areas, reflecting variations in roughness (Figure 1b). Surface-based radar studies have discovered that a significant portion of the Ka-band signal at nadir returns from the snow-ice interface, compared to the typical assumptions of satellite-based Ka-band sea ice altimetry (Stroeve et al., 2020; Nandan et al., 470    2023; Willatt et al., 2023). If backscatter from the snow-ice interface dominated the satellite nadir Ka-band return too, such that the AltiKa echo leading edge mainly represented the illumination of the snow-ice interface, then the floe waveforms from December 2018 would need to be retracked at the thresholds shown in Figure 9c, i.e. at 60-90%. The thresholds from LARM are much closer to those best-matching IS2 than CS2, indicating that the dominant backscattering elevation is well above the snow-ice interface. However, for a perfect match to IS2 (and what we can assume is a more reliable measurement of the air-475    snow interface elevation (Kwok et al., 2019a)) the AK thresholds should look like Figure 9a, i.e. around 20% over the roughest sea ice areas and >50% in areas of new ice formation where floes are smooth and specular. The spread of thresholds in Figure 9a is wider than estimated from the LARM algorithm.



Figure 10 shows a comparison between estimates for the Ku-band radar freeboard and Ka-band radar or laser freeboard
obtained from OIB and IceBird airborne reference data, and coincident freeboard observations from CryoSat-2 (light purple),
AltiKa (light green), and ICESat-2 (blue), at a scale of 25 km, in April 2019. These are the same reference datasets used for
the snow depth comparison in Section 4.4. Each satellite-airborne comparison shows some scatter, with the CryoSat-2 obser-
vations only having a correlation of 0.59 with the reference data. The correlation with the ICESat-2 observations is higher at
0.85 and the laser altimeter accurately captures grid cells with thicker freeboard from 70-90 cm in the reference data. There
are fewer grid cells with reference data at latitudes <81.5°N to evaluate AltiKa but the satellite radar altimeter also captures
thinner freeboards ∼20-30 cm as well as thicker freeboards >70 cm, with a relatively higher correlation of 0.86 with the OIB
laser freeboards. The distributions of the satellite freeboards match closely to the reference data, but are typically narrower
(Figure 10b). This reflects the truncation of the true freeboard distribution in the satellite measurements, where for instance the
km-scale of the radar footprint or interpolation of the SSHA over 10s km tends the measurement more towards the modal than
mean freeboard, at these scales (Ricker et al., 2014; Landy et al., 2020; Belter et al., 2020). This is also one of the main reasons
why the slopes of the freeboard (Figure 10a) and snow depth comparisons (Figure 8) are <1.

The major impact of the radar waveform retracking algorithm on the ice freeboard retrieval is illustrated in Figure 10a. In
dark green and dark purple, respectively, are the same satellite-airborne comparisons but with AltiKa ice floe observations
retracked with TFMRA (threshold first-maximum retracking algorithm) at 50% amplitude on the waveform leading edge and
CryoSat-2 floe observations retracked with TFMRA at 70% amplitude, as described in Armitage and Ridout (2015). The radar
freeboards shown for each sensor are derived with the methods described in Armitage and Ridout (2015) and Lawrence et al.
(2018). With TFMRA retracking, the CryoSat-2 radar freeboards overestimate the reference data by ∼10 cm, especially over
thinner ice with freeboards <25 cm. A similar positive bias, summarized by the dark purple hexagram, was found by Lawrence
et al. (2018). This freeboard bias could be a geophysical effect, as previously suggested, i.e. attenuation of the Ku-band signal
in brine-wetted snow (King et al., 2015; Nandan et al., 2017; Rösel et al., 2021) or volume and internal interface scattering over
snow on MYI (Ricker et al., 2015). Our analysis here shows it could alternatively be a processing effect, i.e. through the choice
of retracker or other processing step in the freeboard calculation. The correlation between CryoSat-2 TFMRA and reference
freeboards is 0.52, and the slopes of the CryoSat-2 fits with different retrackers are similar, so the bias is the key difference.
With TFMRA, the AltiKa radar freeboards have a correlation of 0.77 with the reference freeboards and underestimate the
reference data, with the bias also increasing for thicker ice (dark green in Figure 8a). A significant negative bias, summarized
by the dark green hexagram, was found by Lawrence et al. (2018), who corrected AltiKa (and CryoSat-2) freeboards for the
detected biases with calibration functions based on radar waveform pulse peakiness. A retracking threshold of 50% produces
a relatively similar result to ICESat-2 over thinner FYI floes (Figure 9a), but the threshold needs to be <30% to accurately
measure the air-snow interface elevation over thicker MYI floes (Figure 9a and b). This variable bias could be a geophysical
effect, as previously suggested, caused by significant penetration of the Ka-band signal into snow (Armitage and Ridout, 2015;
Stroeve et al., 2020) or it could, at least partly, be a processing effect of the choice of retracking algorithm and influence of
surface roughness on the optimal retracking threshold (Figures 1b and 9a) (Lawrence et al., 2018).



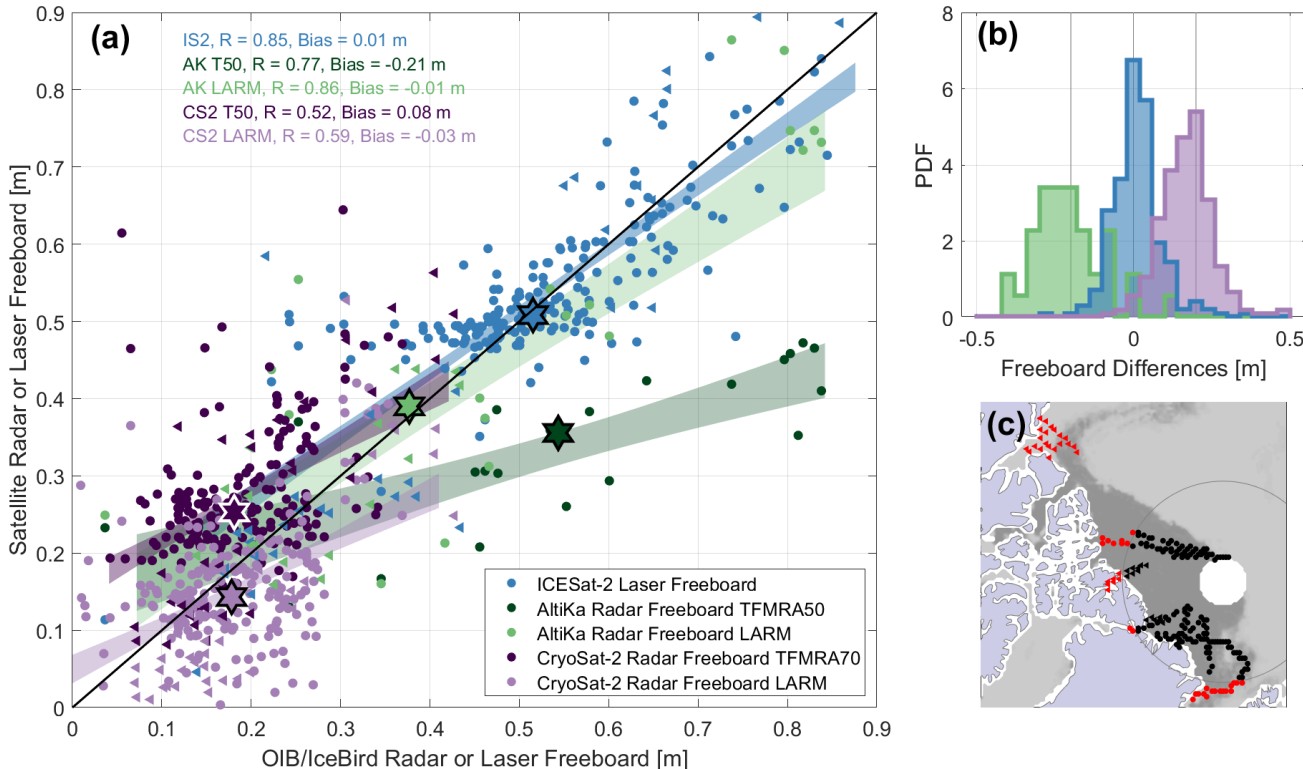

**Figure 10.** Comparison between satellite Ku-band, Ka-band, and laser freeboards, and Ku-band and laser freeboards estimated from airborne data collected by OIB (circles) and IceBird (triangles) in April 2019. (a) Scatterplot of satellite and airborne observations resampled to the same EASE2 grid, with the median of each cluster shown by a hexagram and the 66% confidence interval on the best-fit line through samples shown by the filled envelopes. (b) Distributions of the differences between OIB and IceBird estimated Ku-band radar and measured laser freeboards with corresponding paired satellite freeboards from IS2 (blue), AK LARM (green) and CS2 LARM (purple). The AK distribution has been offset by -0.2 m and the CS2 distribution has been offset by +0.2 m, highlighted by the grey lines. (c) Map of the sea ice type from OSI-403-d, with FYI represented by lighter grey and MYI by darker grey, overlaid with the locations of corresponding airborne and satellite observations above (black) and below (red) the maximum latitude of AltiKa at 81.5°N.



## 5.2 Physical mechanisms behind observed radar scattering biases

It is clear from Figures 5 and 9b that biases remain in the AltiKa radar freeboards, despite improvements that can be made by retracking with a physical model for the radar scattering surface. The assumptions of the physical model, for instance of a single dominant scattering interface, may oversimplify the interactions of the radar with snow and sea ice (Kurtz et al., 2014; Landy et al., 2019). Another possibility is that the large-scale sea ice surface topography ($\sigma$) is not accurately accounted for in the AltiKa scattering model. To explore this, we calculate the mean height differences between AltiKa and ICESat-2 freeboards
as a function of the binned AK $\sigma$ (Figure 11a), obtained from the model inversion (as described in Section 3.3), and backscatter coefficient (Figure 11b), as well as the binned IS2 $\sigma$ (Figure 11c), across the full data record Oct-Apr 2018-2023. This is done separately for 0.1 m intervals of the AK radar or IS2 laser freeboard from 0–0.1 m up to 0.6–0.7 m. For AK freeboards up to 0.4 m (covering 91% of grid cells across the record), the AK-IS2 freeboard bias is approximately independent of the AK surface topography; although AK freeboards <20 cm consistently underestimate those from IS2 (Figure 11a). For AK freeboard >0.4
m, the AK-IS2 freeboard bias depends on surface topography with AK overestimating IS2 by as much as 40 cm when the AK surface topography is very smooth, but only showing a small bias when the AK topography is rougher. These findings are consistent when the bias is analysed as a function of IS2 freeboard and surface topography (Figure 11c), suggesting that the impact of snow surface roughness on the AK retracking point (i.e., Figure 1b) is modelled reasonably well, except in the 3% of cases when AK freeboard is >0.4 m but $\sigma$ is <0.2 m. The high variability of the bias for the roughest IS2 surface topography
(Figure 11c) is caused by limited data: only 2% of IS2 $\sigma$ in the full record are >0.35 m.

The bias between AltiKa and ICESat-2 is always positively sloped for increasing AK backscatter, with a relatively similar gradient between AK freeboard increments. If our assumption of a single dominant scattering surface, i.e., the air-snow interface, was incorrect, and snow-ice interface and/or snow volume scattering were regularly acting to broaden the leading-edge
of the AK waveform producing an overestimated AK $\sigma$, we would expect the AK-IS2 freeboard bias to increase for higher AK $\sigma$. In this case, overestimated AK $\sigma$ would be a strong proxy for over-penetration of the Ka-band radar. Instead, we observe that the bias is largest when there is a strong AK reflection, i.e., high sigma nought (Figure 11b), and smoother AK or IS2 $\sigma$ (Figure 11a). In this scenario (blue lines, representing only ~5% of grid cells across the full record), AK and IS2 $\sigma$ are totally uncorrelated ($r = 0.07$), so the two sensors are likely to be measuring different backscattering surfaces. For the contrasting
scenario, where AK underestimates IS2 (red lines, representing 63% of valid grid cells), the curves are flat so AK $\sigma$ is unlikely to be a proxy for radar penetration (Figure 11a). However, AK freeboard increasingly underestimates IS2 freeboard as the AK backscatter coefficient declines (Figure 11b). This suggests that snow volume scattering (lower sigma nought) may increasingly dominate the returning Ka-band echoes versus surface scattering/reflection (higher sigma nought), as AK increasingly underestimates IS2 freeboard.


Figure 12 shows the derived snow depth difference for KuLa (a) and KuKa (b) between January and April 2019. On average, 8.9 cm of new snow is estimated to have accumulated within the marked area over this period in the KuLa product, whereas



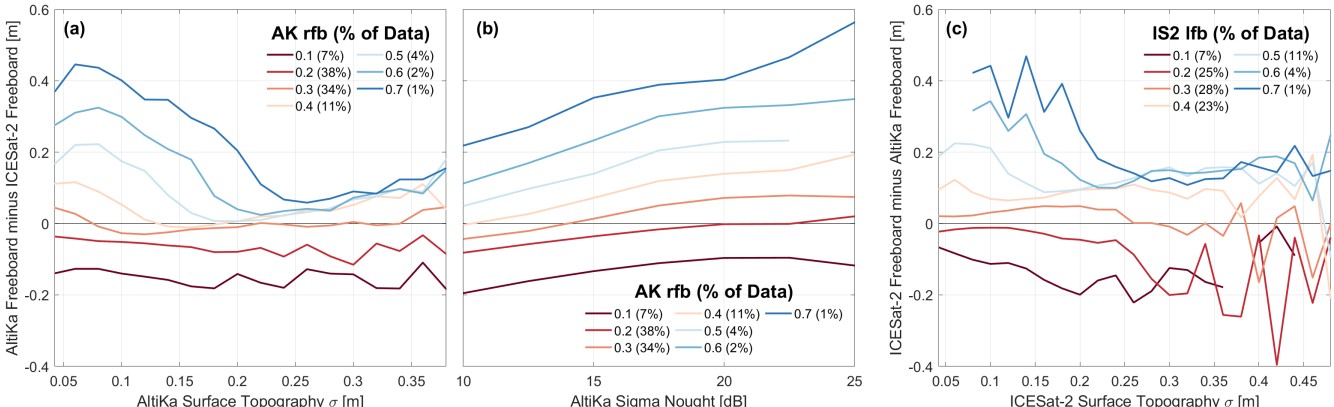

**Figure 11.** Height differences between coincident gridded radar freeboards from AltiKa and laser freeboards from ICESat-2 as a function of (a) the AltiKa radar scattering surface topography ($\sigma_{AK}$), (b) the AltiKa backscatter coefficient (sigma nought), and (c) the ICESat-2 surface topography ($\sigma_{IS2}$), over the period 2018-2023. In (a) and (b) separate lines are shown for 10-cm intervals of the AltiKa radar freeboard and in (c) for 10-cm intervals of the ICESat-2 laser freeboard, up to the value given in the legend. Note the percentages of the data making up each curve in the legends.

only 3.2 cm is estimated to have accumulated in the KuKa product. Over the same period the CS2 large-scale sea ice surface roughness ($\sigma$, derived from LARM) shows a marked increase of 5.9 cm in the same area (Figure 12c). This is typical of the

seasonal roughening of FYI in response to ice deformation (Babb et al., 2020). However, it may be also partly caused by the new snowfall from unusually intense winter storms (Kwok et al., 2020) changing the scattering response of the Ku-band signal so that significant backscatter is sourced from the air-snow interface as well as the snow-ice interface (e.g., Nab et al., 2023), producing an echo with a leading edge including significant contributions from both interfaces (de Rijke-Thomas et al., 2023). Sea ice deformation and increased air-snow scattering can each cause the radar backscattering response to "spread" over a

wider height distribution within the footprint (Landy et al., 2019) and reduce the total backscatter, which indeed drops by 2.6 dB on average (Figure 12d).

In contrast, the AK backscatter barely changes, with an average decrease of only 0.1 dB between January and April (Figure 12e). Although the AK radar freeboard increases by 7.3 cm between January and April, this is around half the IS2 laser

freeboard increase (14.8 cm) and there is no obvious response in the Ka-band backscatter. This suggests that the AK Ka-band signal may only respond weakly to the rapid accumulation of new snow. The Mie scattering coefficient of dry snow at Ka-band approximately halves when the snow density reduces from 350 to 175 kg m$^{-3}$ (Long and Ulaby, 2015). The impact of AK backscatter change on the retracked height has also been recorded as only 5 cm/dB over the Antarctic Ice Sheet (Rémy et al., 2015). Therefore, a lack of backscatter change in Figure 12e suggests the radar remains most sensitive to surface and volume

scattering from snow accumulated earlier in the season, which is deeper in the snowpack by April, and the increase in AK freeboard is mainly due to the ice freeboard thickening rather than new snow accumulation too.







**Figure 12.** (a) KuLa and (b) KuKa snow depth differences [m] between January and April 2019, when a significant amount of new snow is likely to have accumulated on sea ice in the Beaufort, Chukchi and East Siberian Seas. Also shown are the concurrent changes in CryoSat-2 (c)sea ice surface roughness $\sigma$ [m] (derived from LARM) and (d) radar backscatter [dB], and (e) AltiKa radar backscatter [dB]. The grey circles show the limit in coverage for AltiKa, and the black dotted ones mark the approximate area of rapid snow accumulation between January and April.



## 5.3 Snow accumulation over newly-formed sea ice

The basin-wide patterns, seasonal evolution, and validation results of the KuLa snow depths (Figure 8) support the routine operational use of snow depth estimates from CryoSat-2 and ICESat-2. A further useful check on the consistency between these

two sensors is to intercompare the radar and laser freeboards over areas of newly-formed sea ice, where we would expect – in the absence of significant snow – the freeboards to be very similar. To explore this idea, we defined grid cells with new sea ice where new FYI appeared, on any given day between Oct and Dec, 2018-2022, from the 10-km resolution OSISAF *OSI-403-d* Global Sea Ice Type product (Aaboe et al., 2021). In some cases these grid cells will represent FYI advected from an adjacent grid cell, but during the early ice growth period from Oct to Dec this normally represents new FYI formation exceeding the

product's minimum 30% sea ice concentration (SIC) threshold. We then identified any of these "new ice" grid cells that was crossed by CS2 and IS2, within a 5-day period centred on the date of new ice formation and 12.5-km search radius. (We decided to exclude AltiKa observations, given the limited number of occasions where all three sensors coincide within short time window). We used L2 along-track CS2 radar freeboards from LARM and the daily gridded IS2 laser freeboards from ATL20.

The spread of valid "new ice" grid cells is shown in Figure 13c. As expected, they are confined to typical areas of new sea ice formation in the Arctic fall, for instance in the Beaufort and Chukchi Seas, and the Siberian Shelf Seas (which we call the "Pacific Sector" here, marked in blue), and the Greenland, Barents and Kara Seas, and Baffin Bay (which we call the "Atlantic Sector" here, marked in red). There are clear differences in the derived freeboard and snow depth distributions, between these two sectors, across the 5-year study period. The CS2 radar freeboard distribution has a primary peak at 1.2 cm and secondary

peak at 6.9 cm in the Pacific Sector (Figure 13a). The IS2 laser freeboard distribution has a similar shape, with a primary peak at 7.5 cm and secondary peak at 11 cm. The limited number of IS2 freeboards in ATL20 thinner than 3 cm may be caused by the application of a 50% SIC filter and negative freeboards being set to zero in ATL10, rather than a true absence of the thinnest ice. Filtering out dark leads now also reduces the prevalence of very thin IS2 freeboards (Kwok et al., 2021). The small offset in freeboards in the Pacific Sector produces derived snow depths with a mean of only 3.5 cm (Figure 13d), showing that

CS2 and IS2 are measuring approximately the same backscattering surface over sea ice, in the absence of significant snow. In contrast, the radar and laser freeboard distributions are well separated in the Atlantic Sector, with means of 6.6 and 25.9 cm, respectively (Figure 13b). There is a general pattern with thicker derived snow depths in the Barents Sea, around Svalbard, transitioning eastwards to thinner snow depths in the Kara Sea (Figure 13c). The mean snow depth for new sea ice, within five days of formation, is 14.9 cm in the Atlantic Sector. These results support previous studies that suggest frequent intense

cyclones (polar vortices) deposit considerable snow onto sea ice forming in the Barents Sea, soon after the date of formation (Merkouriadi et al., 2017; Graham et al., 2019). The large spread (10.2 cm standard deviation) in snow depths on new ice in the Atlantic Sector (Figure 13d) also suggests that these heavy fall snowfall conditions are variable between years. On the other hand, the cyclone intensity is much lower in the Pacific Sector (Webster et al., 2019) leading to consistently lower rates of snow deposition on newly-forming ice.





**Figure 13.** Estimated fresh snow depth on newly-formed sea ice between 2018 and 2022, including: CryoSat-2 and ICESat-2 freeboards in locations of newly-formed ice, within the Pacific (a) and Atlantic (b) Sectors of the Arctic, snow depth in locations of newly-formed ice with coinciding observations from the two sensors (c), and derived snow depth distributions in the two sectors (blue = Pacific, red = Atlantic, d). The distributions are fit with Gaussian Mixture Models.



## 5.4 Implications for CRISTAL

Although there are similarities between the radar and laser freeboards obtained by coincident Arctic observations from CryoSat-2, AltiKa and ICESat-2, it is not straightforward how these findings will translate to the measurements expected from CRISTAL. For instance, the LRM observations from AltiKa are strongly sensitive to surface roughness, at scales up to the beam-limited footprint of the radar (∼8 km) (Guerreiro et al., 2017), whereas the delay-Doppler SAR mode Ka-band observations from CRISTAL should be much less sensitive to roughness (Wingham et al., 2006). Here, we use the results obtained in Sections 4.1 and 4.2 as the basis for physically-constrained simulations of the CRISTAL dual-frequency altimeter in delay-Doppler SAR mode (see sensor parameters in Table A1). In contrast to the approach described in Section 3.2, here we model the full backscattering response of the snow cover and sea ice, for the chosen month of April 2019, using: (i) the KuLa product for the snow depth, (ii) the CS2 $\sigma$ estimate as the large-scale roughness of the snow-ice interface topography, (iii) the IS2 $\sigma$ estimate obtained by Duncan and Farrell (2022) as the large-scale roughness of the air-snow interface topography, and (iv) the CS2 $s_{rms}$ estimate as the radar-scale roughness of both snow-ice *and* air-snow interfaces. Volume scattering and extinction in the snowpack are modelled with a Mie scattering scheme depending on snow density, grain radius and temperature, as described in Landy et al. (2019). To account for variability in snow density and grain radius, we use all MicroCT observations ($N = 2,846$) collected in the month of April during the MOSAiC field campaign (Macfarlane et al., 2023).

Example Ku- and Ka-band waveforms are modelled for each 25-km grid cell of the April 2019 data (i.e., Figure 5), as the average of 10 echoes simulated with different random realizations of the sea ice topography $\sigma$ and random draws from the MOSAiC snow density and grain radius distributions. FBEM is currently only set up to simulate a single snow layer with a simplified volume scattering scheme that neglects the dense media effects (Tsang et al., 2007), underestimates the complexity of the typical multi-layered snowpack on Arctic sea ice (Macfarlane et al., 2023), and potentially reduces the impact of, for example, volume scattering from relatively large brine-wetted snow grains in the basal depth hoar layer (e.g., Nandan et al., 2017). Using the same $s_{rms}$ value for both interfaces also underestimates the potential for one interface with smoother radar-scale roughness to dominate the total backscatter over the other. However, these simulation results are presented as a first approximation of snow and sea ice echoes that might be expected from CRISTAL. Each simulation includes separate component echoes for the snow surface, snow volume and sea ice surface scattering contributions (Figure 14a and b). For the Ku-band echo, we retrack the total waveform using the threshold determined from only the sea ice surface echo, i.e. following the necessary assumption taken in Section 3.1 as if we did not have snow information available for the simulation. This allows us to estimate the bias in the retracked height of the snow-ice interface, as a fraction of the snow depth, taking into account snow effects on Ku-band scattering (Figure 14a and d). Similarly, for the Ka-band echo, we retrack the total waveform using the threshold determined from only the snow surface echo, allowing us to estimate the bias in the retracked height of the air-snow interface, as a fraction of the snow depth, taking into account Ka-band scattering effects from deeper in the snowpack and potentially from the sea ice surface too (e.g., Willatt et al., 2023) (Figure 14b and d).

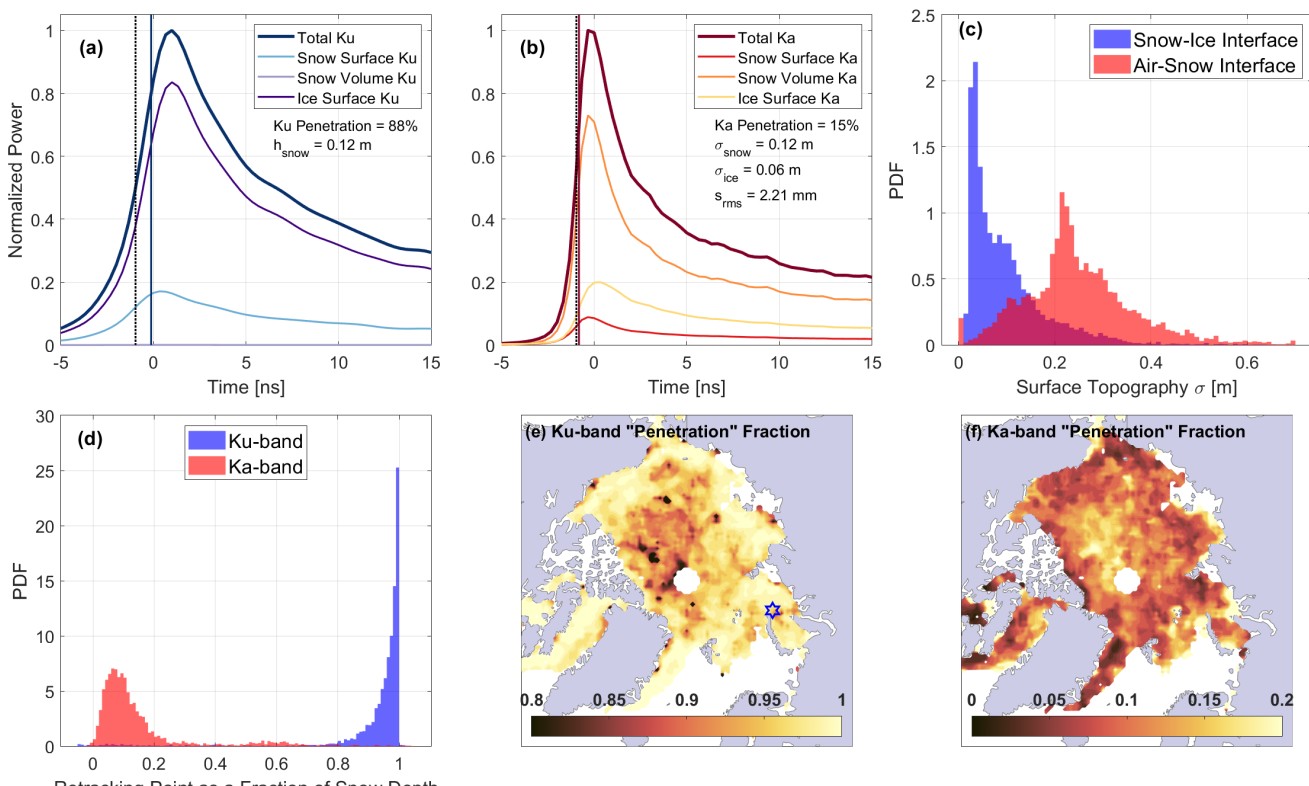

**Figure 14.** Simulated Ku- and Ka-band waveform returns from snow-covered sea ice, based on snow and sea ice geophysical properties estimated in April 2019, for CRISTAL in delay-Doppler SAR mode. (a) and (b) show Ku- and Ka-band component echoes simulated for the grid cell to the north-east of Novaya Zemlya, highlighted with a blue star in (e). The five key parameters listed on (a) and (b) are applicable to both the Ku- and Ka-band simulations. The air-snow interface is identified by the dotted black line, the snow-ice interface by zero time, and the retracking points of the waveforms in blue and red, as described in the text. (c) shows the pan-Arctic distributions of gridded snow-ice ($\sigma_{ice}$) and air-snow interface topography ($\sigma_{snow}$) for the month, as obtained from CS2 and IS2, respectively. (d) shows distributions of the retracking points of Ku- and Ka-band waveforms, as fractions of the snow depth, for all grid cells in the month. (e) and (f) show geographic variations in the "penetration" fraction of the snow depth, where 0 = retracking point at air-snow interface and 1 = retracking point at snow-ice interface.



For the Ku-band returns, the total echoes are typically dominated by the ice surface scattering component (e.g., Figure 14a).
This produces a relatively low bias in the retracking point owing to snow surface and volume scattering effects on the waveform leading edge. The retracked "snow-ice interface" height is located at a median 3% of the relative depth of the snowpack above the true snow-ice interface; however, for one tenth of the simulations the retracked interface is biased >20% of the snow depth above the true ice surface (Figure 14d). Simulations exhibiting a larger bias are typically found in areas of rough MYI (Figure 14e) where the CS2 $\sigma$ and $s_{rms}$ are larger, producing a relatively weaker radar reflection from the snow-ice interface. This
enables volume scattering from snow grains to attenuate the signal and the snow surface reflection to have a larger influence on the waveform leading edge, as shown by Kwok (2014) and de Rijke-Thomas et al. (2023) for airborne Ku-band radar data. At a pan-Arctic scale, the ice surface echo tends to dominate the Ku-band waveform because the slope distribution of the snow-ice interface is significantly smoother than the air-snow interface (Figure 14c). This is backed up by airborne observations, which consistently measured a smoother snow-ice than air-snow interface topography (8 cm mean difference) for the same sea ice
(Landy et al., 2020). Over FYI, the relatively smooth sea ice surface generally produces a strong radar reflection (de Rijke-Thomas et al., 2023) and therefore low bias in the retracking point (Figure 14e).

For the Ka-band returns, the total echoes typically include a strong contribution from snow volume scattering (e.g., Figure 14b). This has been suggested previously for Ka-band returns from snow (Rémy et al., 2015; Guerreiro et al., 2017; Larue
et al., 2021), for instance Adodo et al. (2018) found that the seasonal cycle of Ka-band backscatter over the Antarctic Ice Sheet is governed by the volume echo. However, in the Ka-band waveform simulations of Meloche et al. (2024), the volume backscatter was only found to dominate the total echo when the interface roughness exceeded 3-4 mm. Here, the bias on the retracking point for the air-snow interface is larger than for the snow-ice interface with Ku-band echoes, because a weak reflection from the air-snow interface and volume scattering from deeper within the snowpack, or a strong reflection from the
snow-ice interface, shifts the waveform leading edge to later range bins. The retracked "air-snow interface" height is located at a median 10% of the relative depth of the snowpack below the true air-snow interface; however, for 17% of the simulations the retracked interface is biased >20% of the snow depth below the true snow surface (Figure 14d). The bias is larger when the snow surface scattering contribution is smaller and the volume scattering coefficient is lower (e.g., Rémy et al., 2015), such that backscatter increases less rapidly as the pulse propagates into the snow pack. When the CS2 $\sigma$ is low relative to the IS2 $\sigma$
(Figure 14c), and the snow density and grain size are relatively low, the specular radar reflection from the snow-ice interface can be the strongest contributor to the waveform leading edge and the height of the air-snow interface is underestimated by >50% of the snow depth. In this scenario, the snow-ice interface can occasionally be seen as a secondary peak in the echo, mirroring results from surface-based radar studies at Ka-band (Stroeve et al., 2020; Nandan et al., 2023; Willatt et al., 2023).





## 6   Conclusions


Our study supports the theoretical basis for along-track snow depth on sea ice retrieval from the dual-frequency observations of the CRISTAL mission. The magnitudes of the radar freeboards obtained from the Ku-band SAR mode CryoSat-2 data are significantly thinner than those obtained from the Ka-band LRM mode AltiKa data. However, the freeboards derived from CryoSat-2 and AltiKa observations can vary significantly depending on the techniques used to process them, particularly in the

waveform retracking step. The roughness of the radar scattering surface has a major impact on the threshold that should be used to retrack altimeter waveforms, potentially introducing 10s cm biases into the relative height measured over sea ice floes. Care must therefore be taken in the interpretation of radar freeboards to disentangle the sources of bias coming from geophysical factors, like complex radar scattering over layered snow and sea ice, versus those from processing choices, with respect to some reference "truth". Retracking thresholds of 20-30% over MYI and >50% over new sea ice are required to match AltiKa

radar freeboards to ICESat-2 snow freeboards, covering a wider spread of thresholds than our physical retracker LARM uses to account for surface roughness. This implies that the roughness of the radar scattering surface is not the only factor affecting Ka-band radar freeboards in LRM altimetry.

There are strong similarities in the Arctic basin-wide patterns of freeboard obtained from CryoSat-2, AltiKa, and ICESat-2,

albeit with different magnitudes. Ka-band LRM freeboards can diverge from laser altimeter freeboards when the radar is not sensitive to scattering from newly-deposited snow or when the impacts of surface roughness are not properly accounted for in the freeboard processing. Snow depths estimated from the difference between Ku-band and laser freeboards show accumulation rates of 1.9 cm month$^{-1}$ over FYI and 1.5 cm month$^{-1}$ over MYI; however, the accumulation rates for snow depths estimated from Ka- and Ku-band freeboards were only a third of the KuLa accumulation rates at 0.5 cm month$^{-1}$. The inter-

annual variability of the KuLa snow depths measured over the 5-year record is an order-of-magnitude lower than estimated from the reanalysis-based snow model accumulation scheme SnowModel-LG. Filtering only the cases for which CryoSat-2 and ICESat-2 freeboards observe the same area of newly-forming sea ice, in the MIZ, shows that Pacific Sector snow depths are 3.5 cm within the first 5 days of ice formation whereas Atlantic Sector depths are three times thicker over the same timespan. Despite this rapid initial accumulation, Atlantic Sector snow depths are not considerably thicker than Pacific Sector depths by

the end of the accumulation season in spring.

Model simulations of CRISTAL, in its anticipated sea ice sensing mode, suggest that, in spring, the altimeter will track a median Ka-band elevation 10% below the air-snow interface and a median Ku-band elevation 3% above the snow-ice interface. However, in 17 and 10% of cases, respectively, the retracked elevation is off by more than 20%. Our derived KuKa snow depths have an RMSE of 8.9 cm versus OIB and IceBird airborne observations and 10.4 cm versus the KuLa snow depths (across all

35 measured winter months, from 2018-2023). The CRISTAL mission is required to measure sea ice freeboard, snow depth and ice thickness with uncertainties lower than or equal to 3, 5 and 15 cm, respectively, at a 25-km length scale equivalent to our gridded and airborne and satellite comparisons (Kern et al., 2020). 8-10 cm snow depth uncertainty introduces 17-29



cm sea ice thickness uncertainty (Ricker et al., 2014) even if the Ku-band radar freeboard is measured with absolute certainty.
Consequently, the uncertainties of KuKa snow depths obtained from CryoSat-2 and AltiKa are currently too high to meet the
mission requirements for snow depth and ice thickness uncertainties. However, the novelty of approximately coincident along
track, Doppler-sharpened, Ku and Ka band footprints may enable the snow depth uncertainties estimated from CRISTAL to be
reduced by the 5-6 cm required to meet the mission goals.

Analyzing coincident freeboards and surface roughness observations from three different altimeters, and using these results
to constrain simulations of Ku-/Ka-band SAR-mode radar echoes over snow-covered sea ice, offers some lessons in advance
of the CRISTAL launch. For instance, it would be valuable to study, in more depth: (i) The covariance of the snow and sea
ice surface topography, at scales of 1-1000 m, and its impact on near-nadir radar reflection. (ii) Bounds on mm-cm radar-scale
roughness of snow and ice interfaces and the external factors controlling these roughness variations in time and space. (iii) The
evolution of the satellite Ku- and Ka-band radar return from the same snow and sea ice (e.g., landfast ice) over time to observe
the effects of changing snow properties (depth, grain size, layering, dielectrics) on the backscatter and waveform shape. (iv)
Auxiliary remote sensing observations or model output to support the interpretation of CRISTAL multi-frequency waveforms;
for example, having the anticipated snowpack properties or an estimate for the multi-scale roughness of a radar observation
would enable the application of smart freeboard bias corrections. All these recommendations apply equally to the Southern
Ocean, as much as the Arctic Ocean, where the snow and surface roughness properties can be quite different.

In any case, the CRISTAL observations promise to revolutionize our understanding of the topography and thickness of
Arctic snow and sea ice, as it continues to evolve in response to climate warming, extending the record of sea ice volume
required to understand Arctic mass & energy budgets, improve seasonal ice edge forecasts, and benchmark future climate
model projections.

*Code and data availability.* The *MATLAB* code for simulating CryoSat-2, AltiKa and CRISTAL waveforms is available publicly from https://github.com/jclandy/FBEM. The full 25-km gridded record of CryoSat-2, AltiKa and ICESat-2 freeboards, and derived KuKa and KuLa snow depths, covering the period Oct-Apr 2018-2023, is available from https://doi.org/10.5281/zenodo.13774843

*Author contributions.* JCL conceptualized the study, carried out the main analysis, and wrote the paper. CDRT improved the theoretical
understanding of Ku-band radar interactions with snow and sea ice, and made associated updates to the Facet-based Echo Model. CN, IL and
ARM contributed airborne and in situ reference observations. AAP contributed to the processing of ICESat-2 observations. JCL, CDRT, IL
and MT were involved in the ESA Polar+ Snow Depth on Sea Ice Project that majority supported and led to this work. All authors contributed
to the interpretation of the results, and revised and improved the manuscript.



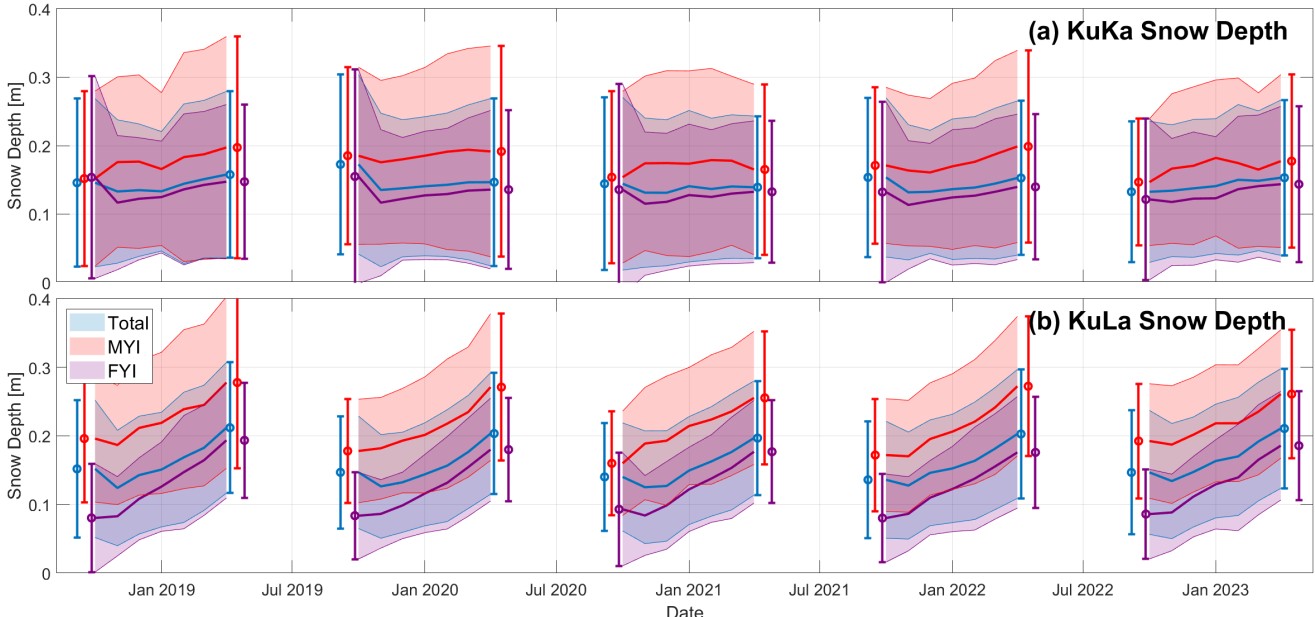

**Figure A1.** Copy of Figure A1 but including KuLa snow depth estimates up to 88°N. Time series for the seasonal change in snow depth obtained from (a) KuKa radar and (b) KuLa radar and laser freeboards over the 2018-2023 sea ice growth/snow accumulation seasons. The envelopes represent +/- one standard deviation around the mean snow depth. The points and whiskers show mean and one standard deviation snow depths at the start and end of each observation season.

*Competing interests.* One of the studies co-authors is on the editorial board of the journal.

*Acknowledgements.* Thanks to Stefan Hendricks (AWI) for compiling the set of gridded reference airborne and *in situ* snow depth observations as part of the ESA Polar+ Snow Depth on Sea Ice project, that are used in Section 4.4.

    Thanks to Michele Scagliola (ESA) and Albert Garcia-Mondejar (isardSAT) for providing the table of expected CRISTAL sensor parameters for the IRIS (Interferometric Radar altimeter for Ice and Snow) instrument operating in sea ice modes, as shown in Table A1.

    JCL, CDRT, IL and MT acknowledge support from the ESA Polar+ Snow Depth on Sea Ice Project (#AO/1-10061/19/I-EF) as the primary
grant supporting this work. JCL, MT and ABF acknowledge additional support from the ESA CLEV2ER (CRISTAL LEVel-2 procEssor prototype and R&D- Sea Ice and Iceberg) project (#AO/1-11448/22/I-AG). JCL further acknowledges support from the INTERAAC (air-snow-ice-ocean INTERactions transforming Atlantic Arctic Climate) project under the Research Council of Norway, RCN (#328957), the SUDARCO (Forskning for god forvaltning av Polhavet) project under the Fram Centre (#2551323), the DynAMIC (Detecting episodes of Arctic sea ice Mass Imbalance) project under RCN (#343069), and the SI/3D (Summer Sea Ice in 3D) project under the European
Research Council, ERC (#101077496). RM and RMFH acknowledges support from the International Space Science Institute team #501. CN acknowledges support from NERC (#NE/S007229/1) and the UK Met Office (CASE Partnership). RMFH further acknowledges support from



**Table A1.** Parameters used for simulations of the Interferometric Radar altimeter for Ice and Snow (IRIS) instrument in delay-Doppler SAR mode at Ku-band and Ka-band frequencies. Hamming weighting is applied prior to the beam-wise azimuth FFT. Other relevant sensor and target parameters not specified here are as listed in Landy et al. (2019).

| Parameter | Unit | Ku-band DD-SAR | Ka-band DD-SAR |
|---|---|---|---|
| Frequency | GHz | 13.5 | 35.75 |
| Bandwidth | MHz | 500 | 500 |
| Satellite Altitude | m | 698900 | 698900 |
| Satellite Velocity | $m\ s^{-1}$ | 7508 | 7508 |
| Pulse-repetition Frequency | Hz | 15119 | 15119 |
| Number of Synthetic Beams | | 64 | 64 |
| Antenna Gain | dB | 42.3 | 50.2 |
| Synthetic Beam Gain | dB | 36.12 | 36.12 |
| Along-track Antenna Parameter | deg | 0.98 | 0.37 |
| Across-track Antenna Parameter | deg | 1.22 | 0.51 |

the Nordic5Tech joint PhD-alliance research project between DTU and NTNU to characterize extreme sea ice features with a combination of remote sensing, in-situ data, and physical modelling, and ESA CRYO2ICEANT 2022 (#4000141420/23/NL/IB/ab).



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
