# Peer review of "Anticipating CRISTAL: An exploration of multi-frequency satellite altimeter snow depth estimates over Arctic sea ice, 2018-2023"

_EGUsphere, 2024_

## Referee Comment (RC2)

Paper review: https://doi.org/10.5194/egusphere-2024-2904

**Anticipating CRISTAL: An exploration of multi-frequency satellite altimeter snow depth estimates over Arctic sea ice, 2018-2023**

Jack C. Landy[1], Claude de Rijke-Thomas[2], Carmen Nab[3,4], Isobel Lawrence[5], Isolde A. Glissenaar[2], Robbie D.C. Mallett[1], Renée M. Fredensborg Hansen[6,7,8], Alek Petty[9], Michel Tsamados[3], Amy R. Macfarlane[1,10], and Anne Braakmann-Folgmann[1]

**General comment**

The study presented in this paper is one of the most advanced analyses of multi-frequency altimetric measurements of sea ice and its snow cover. In a first phase, it provides measurements of the sea ice freeboard using the Ka AltiKa radar altimeter on the Saral satellite, obtained using the LARM physical retracker. This result is already an innovation, as it is the first Ka-band radar freeboard product that provides a realistic topography of sea ice; previous studies focused exclusively on the Ka/Ku differential for snow depth retrieval. This freeboard is then used in combination with the Ku radar freeboard obtained with LARM retracker applied on CryoSat-2 to estimate snow depth. An initial analysis then evaluates this solution against that obtained by combining Ku radar and lidar (KuLa), as well as with airborne snow thickness measurements and the Lagrangian model SnowModelLG. This analysis shows that the snow depth estimate obtained with KuKa seems realistic at the beginning of winter but greatly underestimates this thickness throughout the winter accumulation, even if a slight thickening is observed.

To better understand the origins of this underestimation, an in-depth comparative analysis of each of the freeboards involved in these thickness measurements is provided below, with the aim of better understanding the reasons for these discrepancies: Are they due to retracking problems? to the effects of surface roughness on retracking? to overestimation or underestimation of penetration (or more precisely, variations in the backscatter ratios from air/snow and snow/ice surfaces, and snow volume)?

The results show that it is a combination of these different aspects and allows some of them to be quantified, such as the effect of surface roughness on Ka radar freeboard, which seems to have a negligible effect except in rare situations (high freeboard and low roughness). Nevertheless the Ka freeboard obtained with LARM is underestimated on average relative to IceSat-2, and this is more pronounced for low freeboards aiming to lower snow depth retrieval with KaKu than with LaKu that can reach to only a third of this last one.

The last section presents a simulation of the next CRISTAL dual-frequency altimetry mission which suggests that Ka-band may be underestimated by 10% the total freeboard and the Ku-band overestimated by 3% the ice freeboard. The last section presents a simulation of the upcoming CRISTAL dual-frequency altimetry mission, which suggests that the Ka band could underestimate the total freeboard by 10% and that the

Ku band could overestimate the ice freeboard by 3%. These results are very promising but show that there is still room for improvement. The authors propose various avenues for further study and also emphasize the importance of additional baseline measures.

Given the originality, scientific quality, significance of the implications, and quality of the presentation, I recommend publication of this article with a few minor revisions.

**Detailed remarks**

Lines 19 and 22: In the following sentences, could you clarify what the percentages refer to? *"a median elevation 3% above the snow-ice interface", "median elevation 10% below the air-snow interface"*.

Line 41: CRISTAL will be ready for launch at the end of 2027. This remains the official date for the time being.

Lines 65 and 71: I don't believe that Ku could penetrate 60-90% of the snow depth and Ka 0-40% whatever is the snow depth...

Line 93-95: could be interesting to specify (if possible) from which order of magnitude of altitude the coherent radar reflection becomes dominant for the following analyses; i.e., when going from ground to airborne measurements? or from airborne to space measurements?

Line 148: to be coherent with titles 2.1 and 2.3, the title 2.2 should be: "SARAL AltiKa Observations" as SARAL is the satellite and AltiKa the altimeter.

Line 193: 350kg/m3 for the snow density seems a high value. Could it be justify? Even if the impact is low it could worth to adapt this value.

Line 296: Strange sentence: *With revised classes the waveforms previously classed as ambiguous are now generally classed as sea ice.*

Line 330: Now the snow density varies from 266 to 329 kg/m3 which is not coherent with the previous 350kg/m3 line 193. Could you specify the used speed propagation equation?

Line 496: See last comment (for line 674).

Figure 10a: It's strange to mix-up Ku-band, Ka-band and laser freeboards (both for satellite and airborne)! They should not measure the same surface (air-snow versus snow-ice). Have you applied corrections for the snow impacts (load + speed propagation)? Please justify.

Figure 10b: Which data are used here? LARM? TFMRA? Both? Also it's strange to see a CS2 freeboard greater than IS2 and SRL. The offset is not clearly shown (add arrows?) and it makes the comparisons difficult.

Figure 11: Very interesting plots but Figures 11a and 11c show exactly the opposite

results. I suppose there is an error on the name of the y-axis for 11c (should be SRL-IS2 instead of IS2-SRL).

Line 561: It would be useful to recall here in a short sentence the concept of Mie scattering, as it is very important for understanding the interactions between snow and radar waves.

Line 605: This very important section is not as clear as the previous ones. It could be much clearer if you shortly introduce the objective of the following demonstration instead of just the introductive word 'here'.

Lines 496 and 674: All the analyses regarding the threshold to be used to retrieve coherent results are very interesting but it is important to have in mind that the threshold approach is stable only if its value corresponds to the steepest slope of the waveform leading edge and far from its maximum, i.e. between 30% to 50% as shown Figure 10a in Laforge et al. 2021 https://doi.org/10.1016/j.asr.2020.02.001. For example, in the extreme case of a 100% threshold, this corresponds to take the maximum of the waveform sampled by the altimeter, i.e., a measurement of the epoch on a sampling gate and therefore with a resolution equal to that of the altimeter (about 20 cm for CryoSat-2 SAR). While this does not affect averages over large areas, it does significantly increase the noise in each measurement. Laforge et al. 2021 propose an alternative that involves correcting the range rather than the threshold, as is done for Sea State Bias in the open sea (see Figure 10). I think it is important to keep this alternative in mind. However, retrackers based on a physical model are clearly the best option.

---

## Author Comment (AC2)

**egusphere-2024-2904 Response to Arttu Jutila**

Landy et al.

July 18, 2025

*In this document, we outline our responses to comments, including modifications that we intend to make to the manuscript where necessary. Reviewer comments are shown in black and our responses in blue with intended changes to the revised version of the manuscript given in italics.*

Dear Jack and co-authors,

Kudos to you on your important and timely work on this topic! With this community comment, I would like to raise some points regarding the reference observations (L175ff).

Thank you Arttu for identifying these technicalities and for your comments on the application of pySnowRadar to OIB data. We hadn't considered some of the points you raise, which have prompted us to re-evaluate the OIB snow radar data.

First off, the minor technicalities:

Could you please update the reference Jutila et al. (2021) from the preprint in The Cryosphere Discussions to the published article that has been available now for nearly three years? I mention this here because I have encountered recent papers where this has not always been caught even after professional copy-editing.

Jutila, A., Hendricks, S., Ricker, R., von Albedyll, L., Krumpen, T., and Haas, C.: Retrieval and parameterisation of sea-ice bulk density from airborne multi-sensor measurements, The Cryosphere, 16, 259–275, `https://doi.org/10.5194/tc-16-259-2022`, 2022.

While the geophysical measurement data have not changed, I would appreciate if you would refer to the most recent version (v2) of the AWI IceBird dataset as: Jutila, A., Hendricks, S., Ricker, R., von Albedyll, L., and Haas, C.: Airborne sea ice parameters during the IceBird Winter 2019 campaign in the Arctic Ocean, Version 2 [dataset publication series], PANGAEA, `https://doi.org/10.1594/PANGAEA.966057`, 2024.

Will be corrected.

Then to the more interesting bit, which is applying pySnowRadar and the peakiness method to the April 2019 OIB data.

1. Which flights have you processed exactly?

The snow radar parameter spreadsheet (`snow_param_2019_Greenland_P3.xls` available at `https://gitlab.com/openpolarradar/opr_params`) lists a total of six, not five, flights over sea ice (sheet `\cmd"`, column `\mission_names"`, `\Sea Ice:*"`). Also NSIDC has six files with those dates in

2019 data (`https://doi.org/10.5067/GRIXZ91DE0L9`).

We did not use the flight on April 8th because the flight report included a comment for the snow radar "digital malfunction" and it wasn't used in the OIB-quicklook processing. See `https://daacdata.apps.nsidc.org/pub/DATASETS/ICEBRIDGE/Evaluation_Products/IceBridge_Sea_Ice_Freeboard_SnowDepth_and_Thickness_QuickLook/Documentation/icebridge_ql_products_2019.pdf`. We will clarify this in the manuscript.

2. On L185, you mention using "the same pySnowRadar parameters as the IceBird data", but I guess you mean the peakiness method parameters? pySnowRadar contains also other retrieval algorithm modules like the wavelet method (Newman et al., 2014) with very different parameters for very different purposes.

Yes exactly, will be corrected.

3. From the snow radar parameter spreadsheet notes, it is also obvious that some OIB flights were carried out with a reduced bandwidth (2-8 GHz instead of the full 2-18 GHz) and/or at an unusually high altitude of 3500 ft (nominally ∼1500 ft). The peakiness method was not developed and has not been tested for such missions, and I am curious how the snow depth retrieval results looked like. Did you compare them against the official OIB product at NSIDC (`https://doi.org/10.5067/GRIXZ91DE0L9`)?

We initially used the OIB quicklook product available from NSIDC for all our analyses, but then, based on the issues with the QL method identified in [Kwok et al., 2017], we decided to reprocess the CReSIS data ourselves with the same "peakiness" method applied to the IceBird data, to ensure consistency across the reference data. The reprocessed OIB peakiness data were then used for the snow depth validation and retracker analysis.

There is one case where the flight tracks from OIB on 6th April and from IceBird on 5th April 2019 crossed, with data available from both flights at three of our 25 km grid cells. Both flights were performed with the snow radar covering a 2-18 Ghz bandwidth and at ∼500 m flight altitude. The mean airborne snow depth estimate for each grid cell is shown in Figure 1, including the IceBird peakiness estimate, and OIB quicklook, peakiness and wavelet method estimates. It is evident from these three grid cells, at least, that the peakiness snow depths are very consistent between flights, but the QL and especially wavelet results show thicker snow depths.

To avoid confusing the message, we will not include this analysis in the revised manuscript, but will clarify our method applied to the OIB snow radar data.

4. While its impact is rather minor, which snow density value did you use in the processing? Later, on L193, you mention assuming snow density of 350 kg m-3. Or was it perhaps varying according to Mallett (2024)? In AWI IceBird, it was fixed at 300 kg m-3.

We used a fixed snow density of 300 kg m-3 in our OIB processing for consistency with the IceBird processing, which will now be clarified in the manuscript.

| GRID CELL | LAT | LON | ICEBIRD | QL | PEAKINESS | WAVELET |
|:---:|:---:|:---:|:---:|:---:|:---:|:---:|
| 1 | 83.7330 | -59.7430 | 0.3231 | 0.4350 | 0.3319 | 0.4959 |
| 2 | 84.1190 | -57.5280 | 0.3539 | 0.4175 | 0.3630 | 0.5227 |
| 3 | 84.4940 | -55.0080 | 0.3531 | 0.3953 | 0.3625 | 0.5251 |

Figure 1: Statistics of the snow depths derived from three different processing methods applied to the OIB snow radar data on 6th April 2019, for three 25-km grid cells crossing an IceBird flight on April 5th 2019.

**References**

[Kwok et al., 2017] Kwok, R., Kurtz, N., Brucker, L., Ivanoff, A., Newman, T., Farrell, S., King, J., Howell, S., Webster, M., Paden, J., Leuschen, C., MacGregor, J., Richter-Menge, J., Harbeck, J., and Tschudi, M. (2017). Intercomparison of snow depth retrievals over Arctic sea ice from radar data acquired by Operation IceBridge. *The Cryosphere*, 11:2571–2593,. Type: b.

---

## Author Comment (AC3)

**egusphere-2024-2904 Response to Referee #1**

**Landy et al.**

**July 18, 2025**

In this document, we outline our responses to comments from the reviewer, including modifications that we intend to make to the manuscript where necessary. Reviewer comments are shown in black and our responses in blue with intended changes to the revised version of the manuscript given in italics.

This paper explores existing satellite and airborne radar and laser altimetry observations over the Arctic from 2018-2023 to assess potential future observations from dual frequency Ka and Ku radar from CRISTAL. The study is very well organized, and thoroughly explores similarities and differences in current observations with strong ties to understanding the physical basis for differences. The results are highly impactful and will be an extremely useful reference in preparation for CRISTAL and understanding differences in Ku and Ka radar as well as laser altimetry missions.

I did not see any major technical errors and the explanations and figures were very clear. I just noted some questions that arose while reading the manuscript as outlined below. These are all minor and I would otherwise suggest publication subject to some minor revisions.

Thank you to the reviewer for taking the time to read and check our manuscript, providing valuable comments that improve the clarity of the work.

L118: What is meant by calibrated and uncalibrated observations in this sentence?

This was unclear. We meant the calibrations to OIB freeboards based on radar waveform pulse peakiness, that are applied in the DuST method. Intended revision:

*The DuST (dual-altimeter snow thickness) methodology has more recently been applied to produce pan-Arctic "KuLa" snow depth estimates from the difference between CryoSat-2 observations, with the same calibration to airborne freeboards applied, and ICESat-2 ATL10 observations as part of the...*

L203-204: Can you describe in more detail how the interpolation is done between tie points? Is it linear and over what length scale?

Details on the interpolation method are provided at the end of Section 3.3 and we will now refer to this at the earlier lines, as suggested. The method is linear interpolation then the SSHA profile is smoothed on a 25-km length scale.

[Figure]

Figure 1: CryoSat-2 gridded surface topography vs radar-scale roughness across the entire dataset.

L225-230: Are these four parameter terms independent or are they linked together in some way e.g. the surface topography root-mean square height and mm-cm 'radar-scale' roughness?

No, they are not independent. For instance, the relative position of the tracking point with respect to the waveform peak is related to the surface topography root-mean square height, and the surface topography is indeed related to the mm-cm roughness. A scatterplot between the gridded CryoSat-2 topography and roughness across the entire dataset is shown in Figure 1. There is a nonlinear relationship between the parameters, but there are a significant number of cases where, for example, a large range of mm-cm roughness can be obtained for relatively low surface topography rms height. For clarity, we choose not to go into details on this aspect within the manuscript.

L259: There looks to be a typo here in 95 8%

Thanks, will be corrected.

L270-275: How are the initial starting point values for the lsqnonlin solver determined?

Good question. We will add the following text to the section:

*Initial values for the four free parameters are determined as: $t_0$ at 70% of the waveform leading-edge power, A at the waveform maximum, and $\sigma$ and $s_{rms}$ corresponding to the modelled echo from the lookup table with a peakiness value closest to the peakiness of the observed waveform.*

Figure 2: Can you describe the methodology for discarding secondary peaks? Does this differ between CryoSat-2 and AltiKA?

We will add the following text to the section:

*A filtering routine is applied to exclude samples at major secondary peaks, on the waveform trailing edge, from the model fit (Figure 2). This routine identifies all waveform peaks between the primary peak and noise floor (on the waveform trailing edge) then removes samples if the area of the peak is above a threshold value. The routine is identical between CryoSat-2 and AltiKa, but the thresholds are different.*

L331: Is a consistent snow density as outlined here used also in the processing of the snow radar data?

No it is not. A fixed snow density of 300 kg m$^{-3}$ was applied by [Jutila et al., 2022] to estimate snow depths from the IceBird snow radar observations, so we applied the same fixed snow density to estimate snow depths from the OIB data in April 2019. This was to ensure consistency across the reference snow depth datasets. However, the seasonally-varying function of [Mallett, 2025] used for the satellite data gives a snow density of 329 kg m$^{-3}$ in April, so the difference in the ratio of radar wave velocities in snow and free space would be 0.80 vs 0.79, respectively, between these different snow densities.

L396-400: Can you calculate a skewness for the results? They do indeed appear Gaussian visually, but perhaps this metric could show this quantitatively.

The skewness parameters for Dec 2018 and Apr 2019 are 0.99 and 1.36, so the distributions are positively skewed. The text will be amended to reflect this.

L428: I was confused by the reference to the Beaufort Sea and MOSAiC transects here though see these are discussed a bit later in the paper. The MOSAiC measurements could be discussed in more detail in Section 2 as well.

We will add more detail on the MOSAiC measurements in Section 2.4:

*(ii) Snow depth collected manually with a magnaprobe along the two loops of the MOSAiC campaign Central Observatory transects, approximately weekly [Itkin et al., 2023], accessible from `https://doi.org/10.1594/PANGAEA.937781`, for October 2019 to April 2020.*

At the line specified by the reviewer, we are referring to the thinnest category of snow depths within the reference data compilation described in Section 2.4. Here, the thin snow primarily comes from IceBird airborne observations in the Beaufort Sea in April 2019, and from the MOSAiC transect data in Nov-Dec 2018 when snow only just started to accumulate. Clarification on this will be added to the manuscript.

L588: I'm not sure the statement about filtering out dark leads applies here. Dark leads are only not considered for sea surface height determination, but their freeboard heights still remain in the product.

Based on [Kwok et al., 2021], removal of dark leads from the sea surface height determination produces generally lower height estimates for the sea surface. This would reduce the prevalence of very thin IS2 freeboards regardless of whether the dark lead freeboard heights are retained.

**References**

[Itkin et al., 2023] Itkin, P., Hendricks, S., Webster, M., von Albedyll, L., Arndt, S., Divine, D., Jaggi, M., Oggier, M., Raphael, I., Ricker, R., et al. (2023). Sea ice and snow characteristics from year-long transects at the mosaic central observatory. *Elem Sci Anth*, 11(1):00048.

[Jutila et al., 2022] Jutila, A., Hendricks, S., Ricker, R., von Albedyll, L., Krumpen, T., and Haas, C. (2022). Retrieval and parameterisation of sea-ice bulk density from airborne multi-sensor measurements. *The Cryosphere*, 16(1):259–275.

[Kwok et al., 2021] Kwok, R., Petty, A. A., Bagnardi, M., Kurtz, N. T., Cunningham, G. F., Ivanoff, A., and Kacimi, S. (2021). Refining the sea surface identification approach for determining freeboards in the icesat-2 sea ice products. *The Cryosphere*, 15(2):821–833.

[Mallett, 2025] Mallett, R. D. (2025). A methodologically robust densification function for snow on multiyear arctic sea ice. *Journal of Glaciology*, 71:e24.

---

## Author Comment (AC4)

**egusphere-2024-2904 Response to Referee #2**

Landy et al.

July 18, 2025

In this document, we outline our responses to comments from the reviewer, including modifications that we intend to make to the manuscript where necessary. Reviewer comments are shown in black and our responses in blue with intended changes to the revised version of the manuscript given in italics.

The study presented in this paper is one of the most advanced analyses of multi-frequency altimetric measurements of sea ice and its snow cover. In a first phase, it provides measurements of the sea ice freeboard using the Ka AltiKa radar altimeter on the Saral satellite, obtained using the LARM physical retracker. This result is already an innovation, as it is the first Ka-band radar freeboard product that provides a realistic topography of sea ice; previous studies focused exclusively on the Ka/Ku differential for snow depth retrieval. This freeboard is then used in combination with the Ku radar freeboard obtained with LARM retracker applied on CryoSat-2 to estimate snow depth. An initial analysis then evaluates this solution against that obtained by combining Ku radar and lidar (KuLa), as well as with airborne snow thickness measurements and the Lagrangian model SnowModelLG. This analysis shows that the snow depth estimate obtained with KuKa seems realistic at the beginning of winter but greatly underestimates this thickness throughout the winter accumulation, even if a slight thickening is observed.

To better understand the origins of this underestimation, an in-depth comparative analysis of each of the freeboards involved in these thickness measurements is provided below, with the aim of better understanding the reasons for these discrepancies: Are they due to retracking problems? to the effects of surface roughness on retracking? to overestimation or underestimation of penetration (or more precisely, variations in the backscatter ratios from air/snow and snow/ice surfaces, and snow volume)?

The results show that it is a combination of these different aspects and allows some of them to be quantified, such as the effect of surface roughness on Ka radar freeboard, which seems to have a negligible effect except in rare situations (high freeboard and low roughness). Nevertheless the Ka freeboard obtained with LARM is underestimated on average relative to IceSat-2, and this is more pronounced for low freeboards aiming to lower snow depth retrieval with KaKu than with LaKu that can reach to only a third of this last one.

The last section presents a simulation of the next CRISTAL dual-frequency altimetry mission which suggests that Ka-band may be underestimated by 10% the total freeboard and the Ku-band overestimated by 3% the ice freeboard. These results are very promising but show that there is still room for improvement. The authors propose various avenues for further study and also emphasize the importance of additional baseline measures.

Given the originality, scientific quality, significance of the implications, and quality of the presentation, I recommend publication of this article with a few minor revisions.

Thank you to the reviewer for this nice summary of our manuscript. We're grateful to them for taking the time to check our manuscript, providing valuable comments that improve the clarity of the work.

Lines 19 and 22: In the following sentences, could you clarify what the percentages refer to? "a median elevation 3% above the snow-ice interface", "median elevation 10% below the air-snow interface".

Good catch, this was confusing. We added "..of the snow depth" after the percentages.

Line 41: CRISTAL will be ready for launch at the end of 2027. This remains the official date for the time being.

Yes, the Chief Mission Scientist reminded us of this shortly after the preprint was published ; )

Lines 65 and 71: I don't believe that Ku could penetrate 60-90% of the snow depth and Ka 0-40% whatever is the snow depth...

We suspect that Ku and Ka can penetrate the full snow depth in most scenarios, but the key question is whether the height that returns maximum backscatter corresponds to the snow surface or base, or somewhere in between. If it is somewhere in between, i.e. within the snowpack, then it is more likely the combination of snow surface and basal scattering is producing a mean scattering height within the snowpack than internal snowpack scattering is dominating the return.

Line 93-95: could be interesting to specify (if possible) from which order of magnitude of altitude the coherent radar reflection becomes dominant for the following analyses; i.e., when going from ground to airborne measurements? or from airborne to space measurements?

We cannot reliably estimate whether radar observations from a surface-based or airborne measurement campaign with specific altitude are likely to be dominated by a coherent radar response. However, based on the criteria given in [Fung and Eom, 1983] for the far-field scattering condition, up to a range of a few 100 meters the coherent backscatter at Ku- or Ka-band depends on the range, whereas exceeding this limit it should not. [Fetterer et al., 1992] use a slightly different definition for the one-sided beamwidth of the radar than [Fung and Eom, 1983] and show that incoherent and coherent backscatter coefficients for rough sea ice should be similar at a range of 5-10 km, in Ku-band. [de Rijke-Thomas et al., 2023] found a significant coherent component in OIB Ku-band radar data from relatively smooth landfast sea ice in the Canadian Arctic, when the flight altitude was 500 m.

We can give an order of magnitude of around 1 km for a narrow-beam nadir-looking radar reflection to be dominated by coherent scattering over sea ice, but it should be a future campaign priority to better understand the role of measurement range in coherent vs incoherent near-nadir scattering at Ku- and Ka-band.

Line 148: to be coherent with titles 2.1 and 2.3, the title 2.2 should be: "SARAL AltiKa Observations" as SARAL is the satellite and AltiKa the altimeter.

Good suggestion. Will be corrected.

Line 193: 350kg/m3 for the snow density seems a high value. Could it be justify? Even if the impact is low it could worth to adapt this value.

This is actually an error and will be corrected. We originally used a value of 350 kg m-3, based on the assumption taken in [Lawrence et al., 2018], but then adapted our analysis of the OIB freeboards to use snow radar data processed with the peakiness method rather than quicklook method (as described in the text and in response to questions from Arttu Jutila). For the revised snow radar processing we used a constant value of 300 kg m-3 to be consistent with the IceBird data processing.

Line 296: Strange sentence: With revised classes the waveforms previously classed as ambiguous are now generally classed as sea ice.

We agree and will remove this sentence.

Line 330: Now the snow density varies from 266 to 329 kg/m3 which is not coherent with the previous 350kg/m3 line 193. Could you specify the used speed propagation equation?

As stated in our response to Reviewer 1:
A fixed snow density of 300 kg m$^{-3}$ was applied by [Jutila et al., 2022] to estimate snow depths from the IceBird snow radar observations, so we applied the same fixed snow density to estimate snow depths from the OIB data in April 2019. This was to ensure consistency across the reference snow depth datasets. However, the seasonally-varying function of [Mallett, 2025] used for the satellite data gives a snow density of 329 kg m$^{-3}$ in April, so the difference in the ratio of radar wave velocities in snow and free space would be 0.80 vs 0.79, respectively, between these different snow densities.

The equation for speed-of-light dependence on snow density is Eqn 81 in [Ulaby et al., 1986] Volume 3, page 2061: $c_s = c * (1 + 0.51 * rho_s/1000)^{-1.5}$

The manuscript will be edited to explain our choices on snow density more clearly.

Line 496: See last comment (for line 674).

Figure 10a: It's strange to mix-up Ku-band, Ka-band and laser freeboards (both for satellite and airborne)! They should not measure the same surface (air-snow versus snow-ice). Have you applied corrections for the snow impacts (load + speed propagation)? Please justify.

It was a choice made to simplify the plotting, but all necessary corrections have been applied to compare laser or radar freeboards as "apples to apples".

For instance, the snow depth and speed propagation effect have been taken into account to convert OIB/IceBird laser freeboards and snow radar depths into OIB/IceBird radar freeboards (for comparison with CryoSat-2). No corrections have been applied to the satellite observations here, only to the airborne data so they can be fairly compared to the coinciding satellite freeboard estimates. The full method is described in Section 2.4; however, clarifications will be made to the manuscript where necessary.

Figure 10b: Which data are used here? LARM? TFMRA? Both? Also it's strange to see a CS2 freeboard greater than IS2 and SRL. The offset is not clearly shown (add arrows?) and it makes the comparisons difficult.

The LARM data are used for CryoSat-2 and AltiKa here, as stated in the figure caption. However, there was no consensus on the best way to plot Fig 10b and this was decided as the best option, with the offsets applied to avoid all the distributions overlapping. This option at least shows biases between satellite and airborne data and variability of the paired differences.

Figure 11: Very interesting plots but Figures 11a and 11c show exactly the opposite results. I suppose there is an error on the name of the y-axis for 11c (should be SRL-IS2 instead of IS2-SRL).

We can confirm there is no error on the name of the y-axis. These plots are interesting but challenging to interpret and we spent some time considering them.

It is important to note that the colored lines in Fig 11a and c represent different sets of data: increments of SRL freeboard in a and increments of IS2 freeboard in c. So, if we take the highest freeboard increment, it is not surprising that for locations where SRL freeboards are 60-70 cm the SRL freeboards overestimate IS2, and in the same way for locations where IS2 freeboards are 60-70 cm the IS2 freeboards overestimate SRL. What we took away from these plots is related more to the shapes of the curves, as described in the text. For example, there does not appear to be a relationship between the freeboard bias and surface topography, but there is a relationship between the freeboard bias and SRL backscatter.

Line 561: It would be useful to recall here in a short sentence the concept of Mie scattering, as it is very important for understanding the interactions between snow and radar waves.

Good idea, the following will be added:

*(The Mie scattering coefficient quantifies the scattering of incident EM radiation by particles of similar diameter to the wavelength).*

Line 605: This very important section is not as clear as the previous ones. It could be much clearer if you shortly introduce the objective of the following demonstration instead of just the introductive word 'here'.

We will amend this line to the following:

*To investigate the potential effects of different snow properties and surface roughness on CRISTAL observations, we here use the results obtained in Sections 4.1 and 4.2 as the basis for physically-constrained simulations of the CRISTAL dual-frequency altimeter in delay-Doppler SAR mode..*

Lines 496 and 674: All the analyses regarding the threshold to be used to retrieve coherent results are very interesting but it is important to have in mind that the threshold approach is stable only if its value corresponds to the steepest slope of the waveform leading edge and far

from its maximum, i.e. between 30% to 50% as shown Figure 10a in Laforge et al. 2021 url-https://doi.org/10.1016/j.asr.2020.02.001. For example, in the extreme case of a 100% threshold, this corresponds to take the maximum of the waveform sampled by the altimeter, i.e., a measurement of the epoch on a sampling gate and therefore with a resolution equal to that of the altimeter (about 20 cm for CryoSat-2 SAR). While this does not affect averages over large areas, it does significantly increase the noise in each measurement. Laforge et al. 2021 propose an alternative that involves correcting the range rather than the threshold, as is done for Sea State Bias in the open sea (see Figure 10). I think it is important to keep this alternative in mind. However, retrackers based on a physical model are clearly the best option.

We agree with the analysis and think that such an approach could also be applied in the case of a physical retracker. The physical model results (i.e., a lookup table of modelled echoes) could be used to determine a range correction that depends on certain waveform shape parameters – parameters that can be obtained from both the modelled and observed waveforms. The range correction can be generated relative to a fixed retracking threshold on the steepest, most stable part of the waveform, as suggested.

In any case, here all analyses of freeboards from a threshold retracker are made at the 25-km grid cell scale, integrating typically many hundreds of individual along-track samples, so the impacts of noise in the threshold retracker results should be negligible.

**References**

[de Rijke-Thomas et al., 2023] de Rijke-Thomas, C., Landy, J., Mallett, R., Willatt, R., Tsamados, M., and King, J. (2023). Airborne investigation of quasi-specular ku-band radar scattering for satellite altimetry over snow-covered arctic sea ice. *IEEE Transactions on Geoscience and Remote Sensing*.

[Fetterer et al., 1992] Fetterer, F., Drinkwater, M., Jezek, K., Laxon, S., and Onstott, R. (1992). Sea ice altimetry. In Carsey, F., editor, *Microwave Remote Sensing of Sea Ice (Geophysical Monograph*, pages 111–135. American Geophysical Union, Washington D.C, 62 edition.

[Fung and Eom, 1983] Fung, A. and Eom, H. (1983). Coherent scattering of a spherical wave from an irregular surface. *IEEE Transactions on Antennas and Propagation*, 31(1):68–72.

[Jutila et al., 2022] Jutila, A., Hendricks, S., Ricker, R., von Albedyll, L., Krumpen, T., and Haas, C. (2022). Retrieval and parameterisation of sea-ice bulk density from airborne multi-sensor measurements. *The Cryosphere*, 16(1):259–275.

[Lawrence et al., 2018] Lawrence, I., Tsamados, M., Stroeve, J., Armitage, T., and Ridout, A. (2018). Estimating snow depth over Arctic sea ice from calibrated dual-frequency radar freeboards. *Cryosphere*, 12:3551–3564.

[Mallett, 2025] Mallett, R. D. (2025). A methodologically robust densification function for snow on multiyear arctic sea ice. *Journal of Glaciology*, 71:e24.

[Ulaby et al., 1986] Ulaby, F., Moore, R., and Fung, A. (1986). K.: Microwave remote sensing: Active and passive. Pages: 997 , Volume: 3.